# Variation in ubiquitin system genes creates substrate-specific effects on proteasomal protein degradation

**Mahlon A Collins\*, Gemechu Mekonnen†, Frank Wolfgang Albert\***

Department of Genetics, Cell Biology, and Development, University of Minnesota, Minneapolis, United States

**Abstract** Precise control of protein degradation is critical for life, yet how natural genetic variation affects this essential process is largely unknown. Here, we developed a statistically powerful mapping approach to characterize how genetic variation affects protein degradation by the ubiquitin-proteasome system (UPS). Using the yeast *Saccharomyces cerevisiae*, we systematically mapped genetic influences on the N-end rule, a UPS pathway in which protein N-terminal amino acids function as degradation-promoting signals. Across all 20 possible N-terminal amino acids, we identified 149 genomic loci that influence UPS activity, many of which had pathway- or substrate-specific effects. Fine-mapping of four loci identified multiple causal variants in each of four ubiquitin system genes whose products process (*NTA1*), recognize (*UBR1* and *DOA10*), and ubiquitinate (*UBC6*) cellular proteins. A *cis*-acting promoter variant that modulates UPS activity by altering *UBR1* expression alters the abundance of 36 proteins without affecting levels of the corresponding mRNA transcripts. Our results reveal a complex genetic basis of variation in UPS activity.

**\*For correspondence:**
mahlon@umn.edu (MAC);
falbert@umn.edu (FWA)

**Present address:** †Department of Biology, Johns Hopkins University, Baltimore, United States

**Competing interest:** The authors declare that no competing interests exist.

## Editor's evaluation

The authors use an elegant experimental design to study genetic variation in the ubiquitin-proteasome degradation system in yeast. They identify a large number of QTLs for naturally occurring variation, and they elucidate the causal variants and likely functional mechanisms of several of these. The paper illustrates an innovative new approach to high-throughput QTL mapping for specific molecular processes.

## Introduction

Protein degradation is an essential biological process that occurs continuously throughout the life of a cell. Degradative protein turnover helps maintain protein homeostasis by regulating protein abundance and eliminating misfolded and damaged proteins from cells (*Varshavsky, 2011*; *Collins and Goldberg, 2017*; *Hanna and Finley, 2007*). In eukaryotes, most protein degradation occurs through the concerted actions of the ubiquitin system and the proteasome, together known as the ubiquitin-proteasome system (UPS) (*Coux et al., 1996*; *Collins and Goldberg, 2017*; *Hershko and Ciechanover, 1998*; *Bachmair et al., 1986*; *Ciechanover et al., 2000*). Ubiquitin system enzymes bind degradation-promoting signal sequences, termed degrons (*Varshavsky, 1991*), in cellular proteins and mark them for degradation by covalently attaching chains of the small protein ubiquitin (*Bett, 2016*; *Hershko and Ciechanover, 1998*; *Finley et al., 2012*). The proteasome binds poly-ubiquitinated proteins, then processively deubiquitinates, unfolds, and degrades them to small peptides (*Kisselev et al., 1999*). The UPS degrades a wide array of proteins spanning diverse biological functions and subcellular localizations (*Schwanhäusser et al., 2011*; *Kong et al., 2021*; *Christiano et al., 2020*). By controlling

the turnover of a large fraction of the cellular proteome, the UPS regulates numerous aspects of cellular physiology and function, including gene expression, protein homeostasis, cell growth and division, stress responses, and energy metabolism (*Varshavsky, 2011*; *Hershko and Ciechanover, 1998*; *Hanna and Finley, 2007*; *Pohl and Dikic, 2019*).

Because of the central role of UPS protein degradation in regulating protein abundance, variation in UPS activity can influence a variety of cellular and organismal phenotypes (*Varshavsky, 2011*; *Schwartz and Ciechanover, 1999*; *Hanna and Finley, 2007*; *Schmidt and Finley, 2014*). Physiological variation in UPS activity enables cells to respond to changes in their internal and external environments. For example, UPS activity increases when misfolded or oxidatively damaged proteins accumulate, preventing these molecules from damaging the cell (*Sontag et al., 2014*; *Grimm et al., 2012*; *Finley and Prado, 2020*). Conversely, UPS activity decreases during nutrient deprivation, when the energetic demands of UPS protein degradation would be costly to the cell (*Waite et al., 2016*; *Laporte et al., 2008*; *Bajorek et al., 2003*). Variation in UPS activity may also create discrepancies between protein degradation and the proteolytic needs of the cell, leading to adverse phenotypic outcomes. For example, age-related declines in UPS activity exacerbate the accumulation of damaged and misfolded proteins that occurs during aging, compromising protein homeostasis and, in turn, cellular viability (*Stolzing and Grune, 2001*; *Baraibar and Friguet, 2012*; *Shringarpure and Davies, 2002*). Understanding the sources of variation in UPS activity thus has considerable implications for our understanding of the many traits influenced by protein degradation.

A handful of examples have shown that variation in UPS activity can be caused by individual genetic differences. Rare mutations that ablate or diminish the function of ubiquitin system or proteasome genes impair UPS protein degradation and cause a variety of incurable syndromes. For example, nonsense and frameshift mutations in *UBR1*, an E3 ubiquitin ligase that targets proteins for proteasomal degradation, cause the developmental disorder Johanson-Blizzard Syndrome (*Zenker et al., 2005*). Several *UBR1* missense mutations that moderately decrease Ubr1 activity cause less severe forms of Johanson-Blizzard Syndrome (*Hwang et al., 2011*), suggesting a continuum of variant effects on UPS activity, similar to other genetically complex traits. More recently, proteasome gene missense mutations that impair proteasome assembly and reduce proteasome activity were shown to cause the autoimmune disorder proteasome-associated autoinflammatory syndrome (*Brehm et al., 2015*; *Arima et al., 2011*; *Liu et al., 2012*), further establishing individual genetic differences as a potentially important source of variation in UPS activity. Genome-wide association studies have also linked variation in ubiquitin system (*Xia et al., 2014*; *Diskin et al., 2012*) and proteasome genes (*Cho et al., 2011*) to a variety of disorders, but have neither established the individual causal variants nor tested their effects on UPS activity.

Our understanding of how natural genetic variation affects the UPS comes largely from these limited examples, leaving critical knowledge gaps in several key areas. First, a focus on rare, large-effect mutations linked to Mendelian syndromes likely provides a narrow, incomplete view of the genetics of UPS activity. Most traits are genetically complex, shaped by many loci of small effect and few loci of large effect throughout the genome (*Mackay et al., 2009*; *Ehrenreich et al., 2009*), suggesting that variants that completely or largely ablate UPS gene functions represent only one extreme of a continuum of genetic effects on UPS activity. Second, we have virtually no knowledge of how natural variation in non-UPS genes affects UPS activity. Third, variation in UPS activity can differentially affect the degradation of distinct UPS substrates (*Christiano et al., 2020*; *Kong et al., 2021*). Whether genetic effects on UPS activity affect the turnover of distinct proteins consistently or in a substrate-specific manner remains a fundamentally open question. Finally, we do not know how genetic effects on UPS activity influence other traits. For example, many genetic effects on gene expression influence protein levels without altering mRNA abundance for the same gene (*Battle et al., 2015*; *Ghazalpour et al., 2011*; *Mirauta et al., 2020*; *Albert et al., 2014*; *Brion et al., 2020*; *Chick et al., 2016*; *Cenik et al., 2015*; *Abell et al., 2022*; *Foss et al., 2011*). These protein-specific effects could arise through differences in UPS activity, but there have been no efforts to understand how natural variation that alters UPS activity influences global gene expression at the protein and RNA levels.

Technical challenges have precluded a comprehensive view of the genetics of UPS activity. Mapping genetic influences on a trait with high statistical power requires assaying large, genetically diverse populations of thousands of individuals (*Bloom et al., 2013*). At this scale, in vitro biochemical assays of UPS activity are impractical. Several synthetic reporter systems can measure UPS activity

with high-throughput in vivo (*Geffen et al., 2016*; *Yu et al., 2016*; *Yen et al., 2008*). However, these systems use genetically encoded fluorescent proteins coupled to degrons to measure UPS activity. When deployed in genetically diverse populations, their output is likely confounded from genetic effects on reporter expression levels.

Here, we leveraged advances in synthetic reporter design to obtain high-throughput, reporter expression level-independent measurements of UPS activity in millions of live, single cells. We use these measurements to map genetic influences on the N-end rule, a UPS pathway that recognizes degrons in protein N-termini (N-degrons) (*Varshavsky, 1991*) of thousands of endogenous cellular proteins (*Kats et al., 2018*; *Bartel et al., 1990*; *Hwang et al., 2010*; *Varshavsky, 2011*). Different N-degrons are processed by one of two distinct targeting systems (*Figure 1A*), which allowed us to test for potential pathway-specific effects of natural genetic variation on UPS activity. Systematic, statistically powerful genetic mapping revealed the complex, polygenic genetic architecture of UPS activity. Across the set of 20 N-degrons, we identified 149 loci influencing UPS activity, many of which had pathway- or substrate-specific effects. Resolving causal nucleotides at four loci identified regulatory and missense variants in ubiquitin system genes whose products process, recognize, and ubiquitinate cellular proteins. By measuring the effect of a causal variant in the *UBR1* promoter on the transcriptome and proteome, we implicate genetic influences on UPS activity as a potentially prominent source of post-translational variation in gene expression.

## Results
### Single-cell measurements identify heritable variation in UPS activity

To understand how genetic variation influences UPS activity, we focused on the N-end rule, in which a protein's N-terminal amino acid functions as an N-degron that results in a protein's ubiquitination and proteasomal degradation (*Figure 1A*). The UPS N-end rule can be subdivided into the Arg/N-end and Ac/N-end pathways based on the molecular properties and recognition mechanisms of each pathway's constituent N-degrons (*Figure 1A*; *Varshavsky, 2011*). We reasoned that the breadth of degradation signals and recognition mechanisms encompassed in the N-end rule would allow us to identify diverse genetic influences on UPS activity and that the well-characterized effectors of the N-end rule would aid in defining the molecular mechanisms of variant effects. We used a previously described approach (*Varshavsky, 2005*) to generate constructs containing each of the 20 possible N-degrons and appended these sequences to tandem fluorescent timers (TFTs; *Figure 1A*; *Khmelinskii et al., 2012*). TFTs are fusions of a rapidly maturing green fluorescent protein (GFP) and a slower maturing red fluorescent protein (RFP) (*Khmelinskii et al., 2012*; *Khmelinskii and Knop, 2014*). The TFT's output, expressed as the -$\log_2$ RFP / GFP ratio, is directly proportional to its degradation rate and, when fused to N-degrons, measures UPS N-end rule activity (*Kats et al., 2018*; *Kong et al., 2021*; *Khmelinskii et al., 2012*). Because the TFT is expressed as a single protein construct, the output of the TFT is also independent of its expression level (*Kats et al., 2018*; *Khmelinskii et al., 2014*; *Khmelinskii et al., 2012*; *Kong et al., 2021*), enabling its use in genetically diverse populations.

We characterized the performance of our TFTs by measuring their output in yeast strains with gene deletions that alter UPS activity towards N-end rule substrates. As expected, deleting the E3 ubiquitin ligases of the Arg/N-end (*UBR1*) and the Ac/N-end (*DOA10*) pathways specifically stabilized N-degron TFTs from these pathways (corrected $p < 0.05$ vs. the BY strain, *Figure 1B–D*, *Figure 1—figure supplement 1*, *Figure 1—source data 1*). Deleting *RPN4*, which encodes a transcription factor for proteasome genes, reduces proteasome activity (*Xie and Varshavsky, 2001*) and stabilized reporters from both the Arg/N-end and Ac/N-end pathways (corrected $p < 0.05$ vs. the BY strain, *Figure 1B–D*, *Figure 1—figure supplement 1*, *Figure 1—source data 1*). These results show that our TFTs provide sensitive, quantitative, substrate-specific measures of UPS N-end rule activity.

To understand how natural genetic variation influences UPS activity, we compared two genetically divergent *S. cerevisiae* strains, the "BY" laboratory strain and the "RM" vineyard strain (*Ehrenreich et al., 2009*). RM had higher UPS activity than BY for 9 of 12 Arg/N-degrons and 6 of 8 Ac/N-degrons (corrected $p < 0.05$, *Figure 1D*, *Figure 1—figure supplement 1*, *Figure 1—source data 1*). BY had higher UPS activity than RM for the phenylalanine, tryptophan, and tyrosine Arg/N-degrons (corrected $p < 0.05$, *Figure 1D*, *Figure 1—figure supplement 1*, *Figure 1—source data 1*). BY and RM had similar activity towards the methionine and proline Ac/N-degrons (corrected

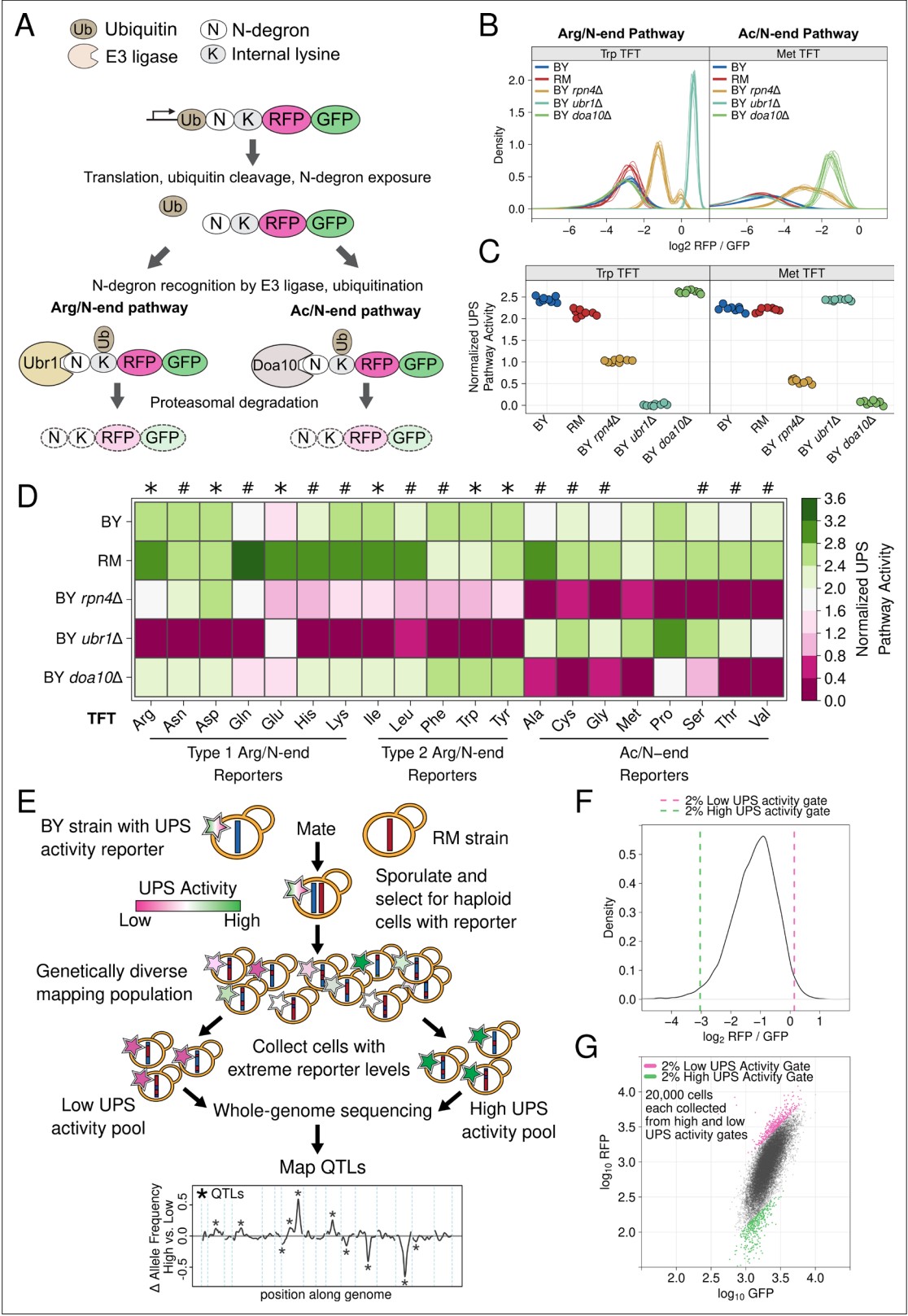

**Figure 1.** UPS N-end rule activity reporters and genetic mapping method. (**A**) Schematic of the production and degradation of UPS activity reporters according to the UPS N-end rule. (**B**) Density plots of the $\log_2$ RFP / GFP ratio from 10,000 cells for each of 8 independent biological replicates per strain per reporter for representative Arg/N-end and Ac/N-end pathway reporters. "BY" and "RM" are genetically divergent yeast strains. "BY $rpn4\Delta$", "BY $ubr1\Delta$", and "BY $doa10\Delta$" carry the indicated gene deletions in the BY background and were used as reporter control strains. (**C**) The median from each

*Figure 1 continued*

biological replicate in B. was scaled, normalized, and plotted as a stripchart such that y axis values are directly proportional to UPS activity. (**D**). Heatmap for all strains and N-degrons using data generated as in C. Symbols above the heatmap denote significant UPS activity differences between BY and RM. "*" indicates 0.05 > Tukey HSD *p* > 1e-6; "#" indicates Tukey HSD *p* < 1e-6. (**E**) Schematic of the bulk segregant analysis genetic mapping method used to identify UPS activity QTLs. (**F**) Density plot of the UPS activity distribution for a genetically diverse mapping population harboring the tryptophan (Trp) N-degron reporter. Dashed vertical lines show the thresholds used to collect cells with extreme UPS activity, which correspond to the high and low UPS activity pools denoted in E. (**G**) Backplot of the cells collected in F. onto a scatter plot of GFP and RFP.

The online version of this article includes the following source data and figure supplement(s) for figure 1:

**Source data 1.** Results of all between-strain comparisons for all N-degron TFTs.

**Figure supplement 1.** Comparison of UPS activity between strains across N-degron reporters.

**Figure supplement 2.** Overview of the constructs and strain construction steps used to generate yeast strains harboring TFT UPS activity reporters.

---

*p* > 0.05, *Figure 1D*, *Figure 1—figure supplement 1*, *Figure 1—source data 1*). Together, these results show that individual genetic differences create heritable, substrate-specific variation in UPS activity.

## Genetic mapping reveals a complex, polygenic genetic architecture for UPS activity

We mapped quantitative trait loci (QTLs) for UPS activity using bulk segregant analysis (*Figure 1E*; *Michelmore et al., 1991*; *Ehrenreich et al., 2010*; *Albert et al., 2014*). In our implementation, this approach attains high statistical power by comparing whole-genome sequence data from pools of thousands of single cells with extreme UPS activity selected from a large population of haploid, recombinant progeny obtained by crossing BY and RM (*Figure 1E–G*; *Ehrenreich et al., 2010*; *Albert et al., 2014*). Using this method, we reproducibly identified 149 UPS activity QTLs across the set of 20 N-degrons at a false discovery rate of 0.5% (*Figure 2A/B*, *Figure 2—source data 1*, *Supplementary file 1*, Appendix 1). The number of QTLs per reporter ranged from 1 (for the Ile N-degron) to 15 (for the Ala N-degron) with a median of 7 (*Figure 2B*, *Figure 2—source data 1*). Using the absolute difference in allele frequency between the high and low UPS activity pools as a measure of effect size, we found that most QTLs had small effects, with only 5 loci (3%) causing an allele frequency difference greater than 0.5 (*Figure 2C*, *Figure 2—source data 1*). Thus, UPS activity is a complex, polygenic trait, shaped by variation throughout the genome.

Analysis of the set of UPS QTLs revealed several patterns. First, the RM allele was associated with higher UPS activity in a significant majority of UPS QTLs (89 out of 149, 60%, binominal test *p* = 0.021, *Figure 2C*), a result that is consistent with our observation that RM had higher UPS activity for 15 of 20 N-degrons (*Figure 1D*, *Figure 1—source data 1*). Second, the number and patterns of QTLs differed between the Ac/N-end and Arg/N-end pathways (*Figure 2B*, *Figure 2—source data 1*). The Ac/N-end pathway was affected by a significantly higher number of QTLs per reporter than the Arg/N-end pathway (9 vs 7, respectively, Wilcoxon test *p* = 0.021), while the QTLs with the largest effect sizes were found for the Arg/N-end pathway (*Figure 2C / D*, *Figure 2—source data 1*).

Third, multiple QTLs for distinct N-degrons occurred in close proximity and had the same direction of effect (*Figure 2B*), suggesting these QTLs may result from the same causal genes or variants. To better understand potential pleiotropy among the set of UPS activity QTLs, we computed overlap among the set of 149 UPS activity QTLs. We considered QTLs for distinct N-degrons overlapping when their peak position occurred within 100 kb and they had the same direction of effect (the sign of the RM allele frequency between the high and low UPS activity pools). Applying these criteria revealed that the 149 UPS activity QTLs were located at 35 distinct QTL regions (*Figure 2—source data 2*). Of these 35 regions, 23 (66%) affected only reporters from either the Arg/N-end (12) or Ac/N-end (11) pathways of the N-end rule (*Figure 2—source data 2*). Five of the 23 pathway-specific QTL regions affected only individual N-degrons (*Figure 2—source data 2*). Use of more lenient LOD score thresholds for QTL detection did not alter these general conclusions (*Figure 2—source data 2*, *Supplementary file 2*). Thus, the majority of QTLs for the N-end rule are pathway-specific, revealing considerable complexity in the genetics of UPS protein degradation.

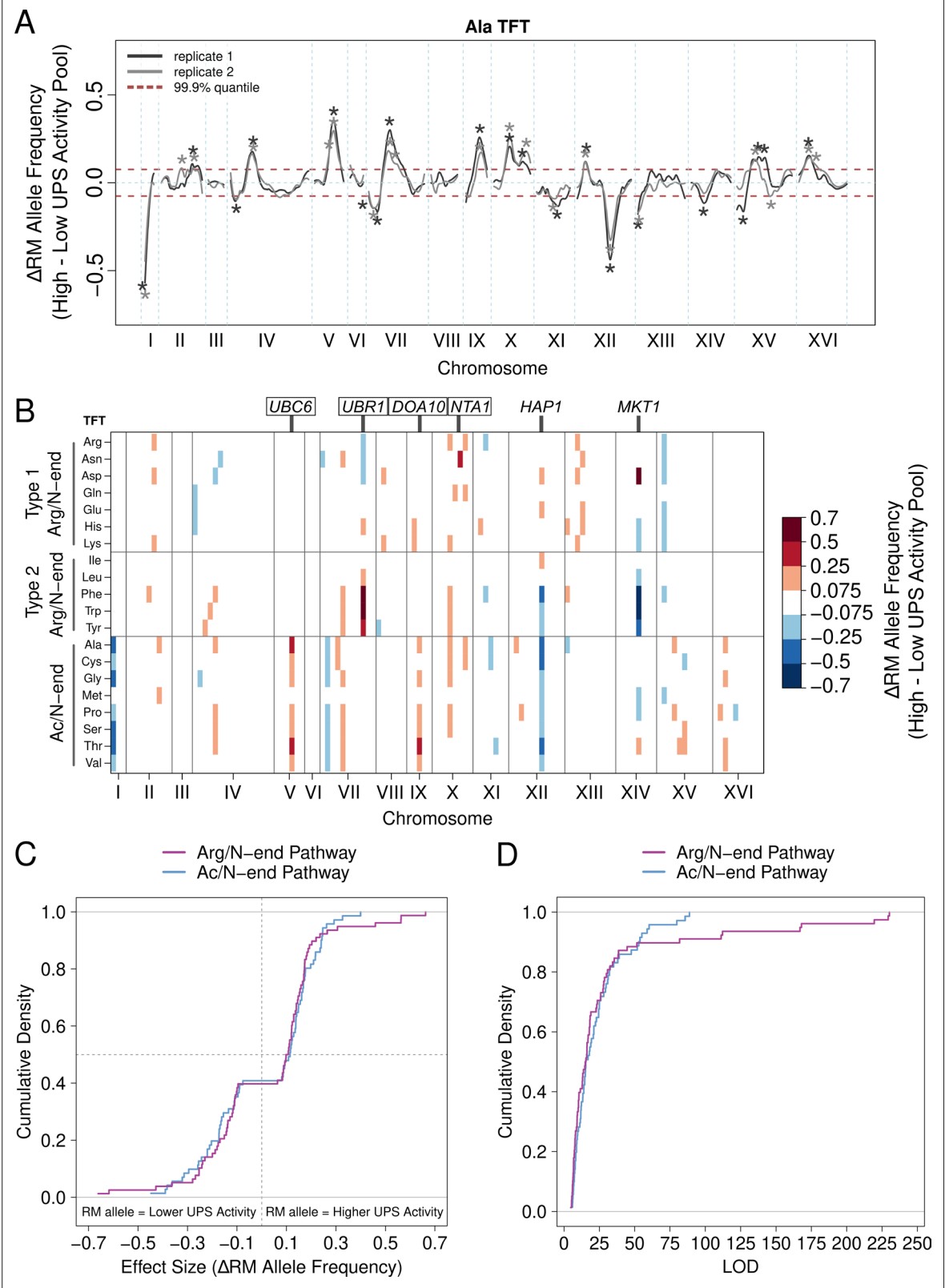

**Figure 2.** UPS activity QTL mapping results. (**A**) Results from the alanine (Ala) N-degron reporter illustrate the results and reproducibility of the method. Asterisks denote QTLs, colored by biological replicate. (**B**) QTL mapping results for the 20 N-degrons. Colored blocks of 100 kb denote QTLs detected in each of two independent biological replicates, colored according to the direction and magnitude of the effect size (RM allele frequency difference between high and low UPS activity pools). Experimentally validated (boxed) and candidate (unboxed) causal genes for select QTLs are annotated above

*Figure 2 continued on next page*

*Figure 2 continued*

the plot. (**C**) Cumulative distributions of the effect size and direction for Arg/N-end and Ac/N-end QTLs. (**D**) Cumulative distribution of LOD scores for Arg/N-end and Ac/N-end QTLs.

The online version of this article includes the following source data for figure 2:

**Source data 1.** All N-end rule QTLs.

**Source data 2.** All distinct N-end rule QTL regions.

## Multiple causal DNA variants in *UBR1* create substrate-specific effects on UPS activity

We leveraged the high degree of pathway specificity in our N-end rule QTLs to aid in the identification of causal genes in broad genomic QTL regions. A QTL on chromosome VII detected with 8 of 12 Arg/N-degron reporters (*Figure 2B*) was centered on *UBR1*, the E3 ligase that recognizes Arg/N-degrons and targets them for UPS protein degradation (*Figure 3A*). To determine whether *UBR1* contains causal DNA variants for UPS activity towards Arg/N-degrons, we used the CRISPR-swap allelic engineering method (*Lutz et al., 2019*) to create BY strains with RM *UBR1* alleles (see 'Materials and methods'). Arg/N-degrons are classified as Type 1 or 2 depending on their Ubr1 binding site (*Varshavsky, 2011*; *Bartel et al., 1990*). The RM allele at the chromosome VII QTL was associated with decreased UPS activity towards Type 1 Arg/N-degrons and increased UPS activity towards Type 2 Arg/N-degrons (*Figure 2B*). We therefore tested the effects of the RM *UBR1* alleles on two Type 1 (asparagine [Asn] and aspartate [Asp]) and two Type 2 Arg/N-degrons (tryptophan [Trp] and phenylalanine [Phe]).

Consistent with our QTL mapping results, The RM *UBR1* allele significantly decreased the degradation rate of Type 1 Arg/N-degrons and increased the degradation rate of Type 2 Arg/N-degrons (corrected $p < 0.05$, *Figure 3B/C*, *Figure 3—figure supplement 1*). Thus, *UBR1* is a causal gene for the chromosome VII QTL, and BY / RM variants in *UBR1* differentially affect the degradation of Type 1 and 2 substrates of the Arg/N-end pathway.

QTL causal genes may contain multiple causal variants, making it necessary to test the effects of individual gene regions and variants in isolation (*Lutz et al., 2019*; *Abell et al., 2022*; *Laurie-Ahlberg and Stam, 1987*). We used CRISPR-swap to test the effect of partial RM *UBR1* alleles on UPS activity towards Type 1 Arg/N-degrons. The RM open-reading frame (ORF) significantly decreased the degradation of the Asn, but not the Asp TFT (*Figure 3C*, *Figure 3—figure supplement 1*). The RM *UBR1* promoter and terminator did not affect UPS activity towards either reporter (corrected $p > 0.05$, *Figure 3C*, *Figure 3—figure supplement 1*). Thus, variants in the RM *UBR1* ORF are the main determinant of the gene's effects on the Asn N-degron, while the effects of the RM *UBR1* alleles on the Asp N-degron may be driven by epistatic interactions between variants in the promoter, ORF, and terminator.

The partial RM *UBR1* alleles had drastically different effects on the degradation of Type 2 Arg/N-degrons (*Figure 3C*). Both the RM *UBR1* promoter and ORF significantly increased UPS activity towards the Type 2 Trp and Phe Arg/N-degrons (corrected $p < 0.05$, *Figure 3B/C*, *Figure 3—figure supplement 1*). The RM *UBR1* terminator did not affect the degradation of either Type 2 Arg/N-degron (corrected $p > 0.05$, *Figure 3B / C*, *Figure 3—figure supplement 1*). Thus, the RM *UBR1* promoter and ORF each contain at least one causal variant that increases UPS activity towards Type 2 Arg/N-degron-containing substrates. Together with our Type 1 Arg/N-degron fine-mapping, these results establish that UPS activity QTLs can contain multiple causal DNA variants in a single gene that can differentially affect the turnover of distinct UPS substrates.

To identify individual causal variants, we tested the effect of the two BY / RM *UBR1* promoter variants (*Figure 3D*) on UPS activity towards Type 2 Arg/N-degrons. The –469A>T variant significantly increased the degradation rate of the Trp and Phe N-degrons (corrected $p < 0.05$, *Figure 3E*, *Figure 3—figure supplement 2*). By contrast, the –197T>G variant had no effect on either N-degron, establishing –469A>T as the causal nucleotide in the *UBR1* promoter (corrected $p > 0.05$, *Figure 3E*, *Figure 3—figure supplement 2*). The magnitude of the effect caused by –469A>T suggests that this variant accounts for the majority of *UBR1* effects on the degradation of Type 2 Arg/N-end substrates (*Figure 3B/C/E*, *Figure 3—figure supplements 1 and 2*).

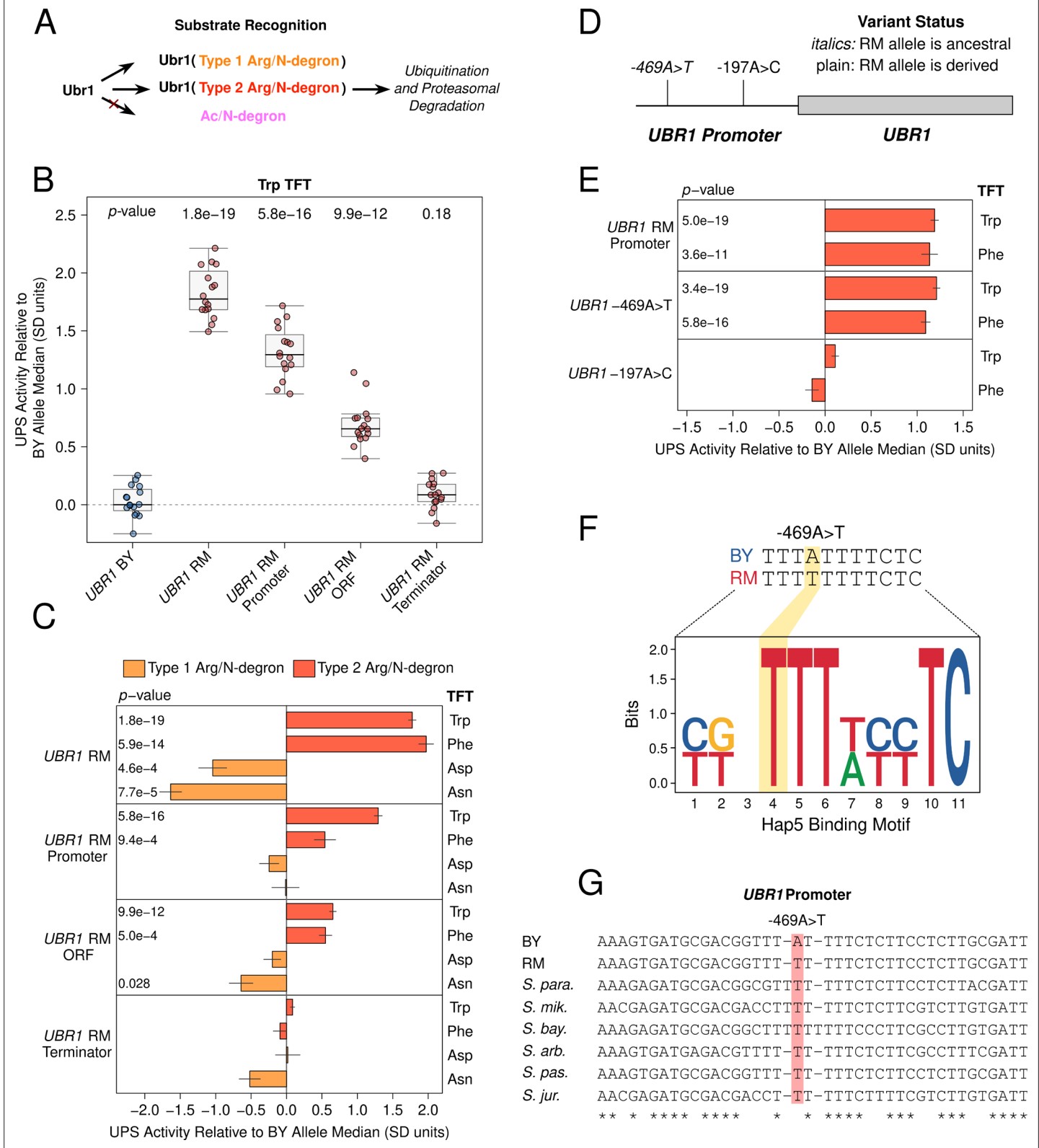

**Figure 3.** Substrate-specific effects of *UBR1* variants on the degradation of Arg/N-degrons. (**A**) Schematic illustrating Ubr1's role in Arg/N-degron recognition. (**B**) Multiple causal DNA variants in *UBR1* shape UPS activity towards the Trp N-degron. The BY strain was engineered to contain full or partial RM *UBR1* alleles as indicated and UPS activity towards the Trp N-degron TFT was measured by flow cytometry. UPS activity was Z-score normalized and scaled relative to the median of a control BY strain engineered to contain the full BY *UBR1* allele. Each point in the plot shows the

*Figure 3 continued on next page*

*Figure 3 continued*

median of 10,000 cells for each of 16 independent biological replicates per strain. *p*-values at the top of the plot display the Benjamini-Hochberg corrected *p*-value for the t-test of the indicated strain versus the strain with the BY *UBR1* allele. Box plot center lines, box boundaries, and whiskers display the median, interquartile range, and 1.5 times the interquartile range, respectively. (C). Barchart summarizing the effects of RM *UBR1* alleles on UPS activity towards the indicated Type 1 and 2 Arg/N-degrons using data generated as in B. *p*-values in the plot display the Benjamini-Hochberg corrected *p*-value for the t-test of the indicated strain versus the control strain engineered to contain the BY *UBR1* allele. (D) Diagram of the individual BY / RM *UBR1* promoter variants. (E) as in C., but for the RM *UBR1* promoter and individual BY / RM *UBR1* promoter variants. (F) Sequence logo of the Hap5 binding motif created by the causal –469A>T *UBR1* promoter variant. (G) Multi-species alignment of the *UBR1* promoter at the causal –469A>T variant. Abbreviations: '*S. para.*', Saccharomyces paradoxus; '*S. mik.*', Saccharomyces mikatae; '*S. bay.*', Saccharomyces bayanus; '*S. arb*', Saccharomyces arboricola; '*S. pas.*', Saccharomyces pastorianus; '*S. jur*', Saccharomyces jurei.

The online version of this article includes the following figure supplement(s) for figure 3:

**Figure supplement 1.** Raw *UBR1* full gene fine-mapping data.

**Figure supplement 2.** Raw *UBR1* promoter fine-mapping data.

**Figure supplement 3.** Population frequency and distribution of the causal *UBR1* –469A>T variant.

To gain further insight into the causal –469A>T variant, we examined its molecular properties, evolutionary history and population frequency using genome sequence data from a panel of 1,011 *S. cerevisiae* strains (*Peter et al., 2018*). The BY allele of the causal –469A>T variant in the *UBR1* promoter disrupts a predicted binding site for the transcription activator Hap5 (*Figure 3F*) and decreased the output of a synthetic reporter in a massively parallel study of yeast promoter variants (*Renganaath et al., 2020*). BY carries the derived 'A' allele at –469A>T, which occurs in a poly(T) motif that is highly conserved across yeast species (*Figure 3G*). The population frequency of –469A>T is 1% and the variant is found in only in the Mosaic Region 1 clade that contains the BY strain (*Figure 3—figure supplement 3*). These results suggest the BY allele decreases UPS activity by decreasing *UBR1* expression, which we subsequently validated at the RNA and protein levels (Figure 5). Moreover, the derived status and low population frequency of the BY allele at position –469 suggests that it may negatively impact organismal fitness, a notion consistent with the generally deleterious consequences of reduced *UBR1* activity or expression (*Zenker et al., 2006*; *Zenker et al., 2005*; *Chen et al., 2006*).

## Causal variants in functionally diverse ubiquitin system genes influence UPS activity

Some of the QTLs with the largest effects were specific to distinct N-end rule pathways or substrates and centered on known ubiquitin system genes (*Figure 2B*). We used allelic engineering to test whether these genes contained causal DNA variants for UPS activity.

A QTL on chromosome X was specific to the Type 1 asparagine (Asn) N-degron of the Arg/N-end pathway (*Figure 2B*). The QTL's peak occurred within *NTA1*, which encodes an amidase that converts N-terminal asparagine and glutamine residues to aspartate and glutamate, respectively (*Figure 4A*). This processing is necessary to convert Asn and Gln N-ends into functional N-degrons (*Baker and Varshavsky, 1995*). *NTA1* contains multiple BY / RM promoter variants and two missense variants that alter amino acids on the protein's exterior surface (*Figure 4B/D*). Consistent with the chromosome X QTL effect, the full RM *NTA1* allele significantly increased the degradation rate of the Asn TFT (corrected *p* < 0.05, *Figure 4C*, *Figure 4—figure supplement 1*). The RM *NTA1* promoter did not alter the degradation rate of the Asn TFT (corrected *p* > 0.05, *Figure 4C*). Instead, the two BY / RM *NTA1* missense variants, D111E and E129G, both influenced degradation of the Asn TFT, but in opposite directions. D111E decreased the Asn TFT's degradation, while, E129G increased it (corrected *p* < 0.05, *Figure 4C*, *Figure 4—figure supplement 1*). The effect of E129G was in the same direction as that of the chromosome X QTL and was approximately threefold greater than that of the effect of D111E (*Figure 4C*, *Figure 4—figure supplement 1*). Thus, at *NTA1*, one causal variant's large effect masks the opposing, smaller effect of a second causal variant.

A QTL on chromosome IX detected for 6 of 8 Ac/N-end degrons contained *DOA10*, the E3 ligase of the Ac/N-end rule pathway (*Figure 4E*). The effect size of this QTL varied between Ac/N-degrons. We therefore tested the glycine (Gly) and threonine (Thr) reporters to determine whether BY / RM *DOA10* variants exert substrate-specific effects on UPS activity. The RM *DOA10* allele contains three missense variants, Q410E, K1012N, and Y1186F, and does not contain promoter or terminator variants (*Figure 4F/H*). The full RM *DOA10* allele significantly increased the degradation of both the Gly and

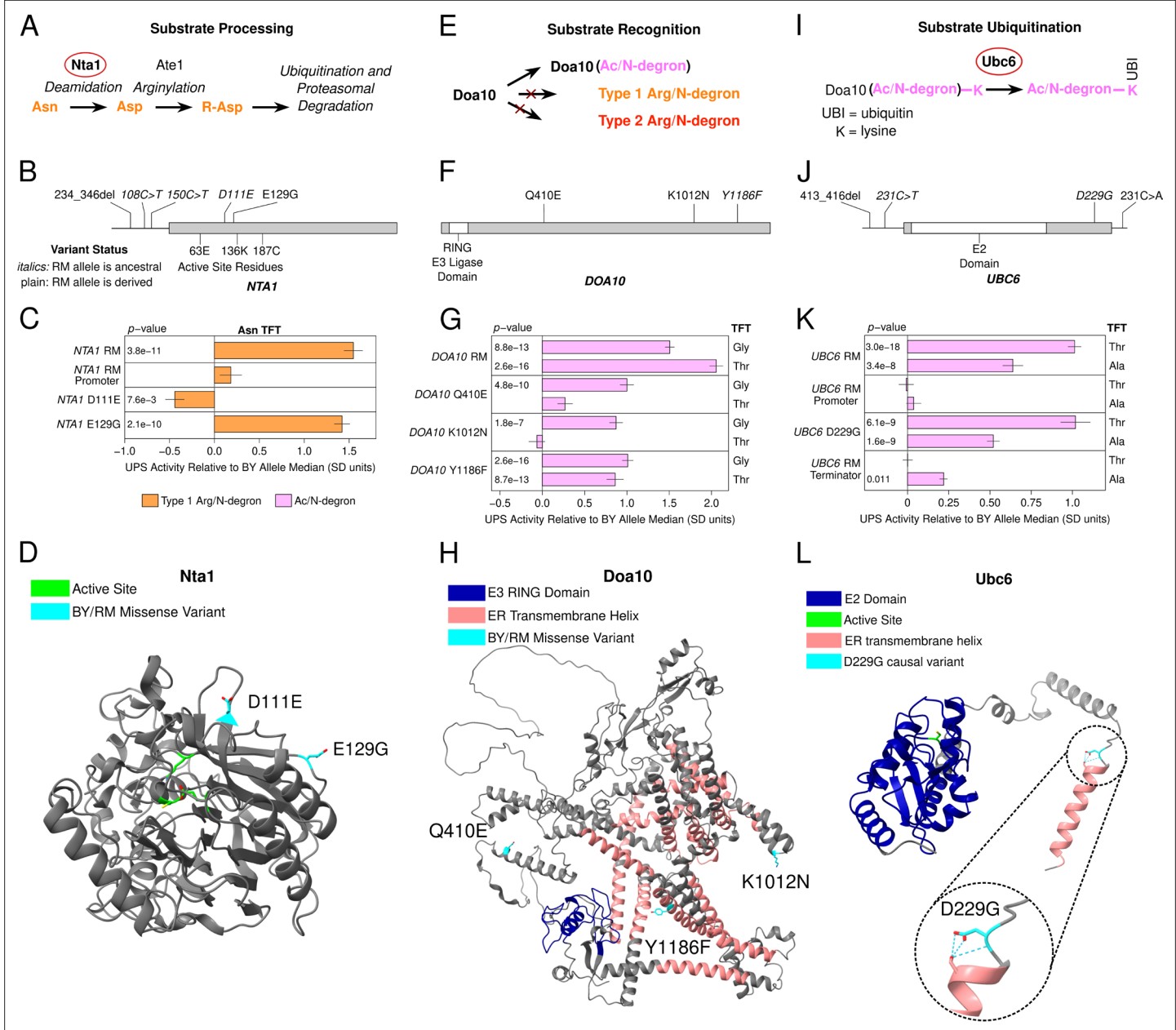

**Figure 4.** Identification of causal DNA variants for UPS activity in functionally diverse ubiquitin system genes. (**A**, **E**, and **I**). Schematics showing the role of Nta1 (**A**), Doa10 (**E**), and Ubc6 (**I**) in UPS substrate processing, recognition, and ubiquitination, respectively. (**B**, **F**, and **J**). Location of regulatory and missense BY / RM variants, as well as active sites and functional domains in the proteins encoded by *NTA1* (**B**), *DOA10* (**F**), and *UBC6* (**J**). **C**., **G**., and **K**. Fine-mapping results for *NTA1* (**C**), *DOA10* (**G**), and *UBC6* (**K**). Benjamini-Hochberg corrected *p*-values are shown for the t-test of the indicated strain versus a control BY strain engineered to contain the BY allele of each gene. AlphaFold predicted protein structures for Nta1 (**D**), Doa10 (**H**), and Ubc6 (**L**) are shown with causal DNA variants, functional domains, active sites, and transmembrane helices highlighted. The inset in L. shows a predicted hydrogen bonding network at residue 229 in the BY Ubc6 protein.

The online version of this article includes the following figure supplement(s) for figure 4:

**Figure supplement 1.** Raw *NTA1* fine-mapping data.

**Figure supplement 2.** Raw *DOA10* fine-mapping data.

**Figure supplement 3.** Raw *UBC6* fine-mapping data.

**Figure supplement 4.** Population frequencies and distributions of causal variants.

Thr TFTs (corrected $p < 0.05$, *Figure 4G*, *Figure 4—figure supplement 2*). When tested in isolation, all three BY / RM *DOA10* missense variants increased the degradation of the Gly TFT (corrected $p < 0.05$, *Figure 4G*, *Figure 4—figure supplement 2*). In contrast, only the Y1186F variant significantly increased the degradation of the Thr TFT (*Figure 4G*, *Figure 4—figure supplement 2*). The multiple causal variants and substrate-specific effects of individual *DOA10* variants further highlights the complex effects of variation in E3 ubiquitin ligases on UPS activity.

A QTL on chromosome V detected for 7 of 8 Ac/N-degrons contained *UBC6*, the E2 ubiquitin-conjugating enzyme of the Ac/N-end pathway. Ubc6 pairs with Doa10 to ubiquitinate substrates of the Ac/N-end pathway (*Figure 4I*; *Chen et al., 1993*; *Sommer and Jentsch, 1993*; *Swanson et al., 2001*). The RM *UBC6* allele contains a 3 base pair deletion in the promoter, one missense variant, and one terminator variant (*Figure 4F*). Using the threonine (Thr) and alanine (Ala) TFTs, we established that *UBC6* is a causal gene for this QTL (corrected $p < 0.05$, *Figure 4K*, *Figure 4—figure supplement 3*). The D229G missense variant altered UPS activity towards the alanine (Ala) and threonine (Thr) Ac/N-degrons (corrected $p < 0.05$, *Figure 4K*, *Figure 4—figure supplement 3*). The *UBC6* RM terminator also significantly increased the degradation rate of the Ala, but not Thr TFT, further establishing the substrate-specific effects of genetic variation on UPS activity (*Figure 4K*, *Figure 4—figure supplement 3*). Our results with *UBC6* show that genetic variation influences UPS activity through effects on substrate ubiquitination by E2 ubiquitin conjugating enzymes, as well as substrate recognition by the E3 ligases Ubr1 and Doa10 described above.

Knowledge of the causal nucleotides in *NTA1*, *DOA10*, and *UBC6* allowed us to examine their molecular properties, evolutionary histories, and population frequencies. A notable feature of causal missense variants was the distal location of their encoded amino acids relative to the active site of the corresponding protein (*Figure 4D/H/L*). The amino acids encoded by the *NTA1* causal variants occur on the protein's exterior surface (*Figure 4D*), while those for the *DOA10* and *UBC6* causal variants occur in or near transmembrane helices that anchor these proteins to the endoplasmic reticulum membrane (*Figure 4H/L*). Thus, in addition to effects on ubiquitin system gene expression, such as with *UBR1* –469A>T, the molecular basis for the continuous distribution of variant effects on UPS activity may also involve subtle alterations to the stability, localization, or physical interactions of ubiquitin system proteins (*Oh et al., 2020*).

The BY alleles of the causal *DOA10* Y1186F and *UBC6* D229G variants are each derived, at low population frequencies (5.1% and 2.2%, respectively), and occur in only 3 non-BY clades (*Figure 4—figure supplement 4*), similar to the –469A>T *UBR1* variant. Given the generally deleterious effects of reduced UPS activity (*Pohl and Dikic, 2019*; *Schwartz and Ciechanover, 1999*), these variants may be subject to purifying selection. In contrast, the RM allele of the causal *NTA1* E129G variant is derived, common (51.5% population frequency), and found in most clades (*Figure 4—figure supplement 4*). The derived RM allele of the causal *NTA1* E129G variant may have been able to rise to comparatively high population frequency because deleting *NTA1* does not decrease competitive fitness (*Baker and Varshavsky, 1995*).

We examined additional QTLs to nominate candidate causal genes. The most frequently observed UPS QTL was detected for 8 of 8 Ac/N-end and 6 of 12 Arg/N-end TFTs and was located on chromosome XII in the immediate vicinity of a Ty1 insertion in the *HAP1* transcription factor in the BY strain (*Figure 2B*, *Figure 2—source data 1*; *Gaisne et al., 1999*). The Ty1 insertion in *HAP1* exerts highly pleiotropic effects on gene expression, altering the expression of 3,755 genes (*Albert et al., 2018*). Similarly, a QTL on chromosome XIV affected 10 of our 20 N-degron TFTs and contained the *MKT1* gene (*Figure 2B*, *Figure 2—source data 1*). *MKT1* encodes a multi-functional RNA binding protein involved in the post-transcriptional regulation of gene expression and is the causal gene for other QTLs previously mapped in the BY/RM cross (*Jain et al., 2016*; *Wickner, 1987*; *Icho et al., 1986*). *HAP1* and *MKT1* are the likely causal genes for the chromosome XII and XIV QTLs, showing that genetic variation may also shape UPS activity through indirect effects on genes with no known connection to the UPS.

Taken together, our analysis of causal genes and nucleotides illustrates the breadth and diversity of genetic influences on UPS activity. Each fine-mapped causal gene harbored multiple causal variants that may differentially affect distinct UPS substrates. Regulatory and missense variants in ubiquitin system genes that shape the full sequence of molecular events in protein ubiquitination, including substrate processing, recognition, and ubiquitination, alter UPS activity.

## Protein-specific effects of *UBR1* -469A>T on gene expression

Previous efforts to understand how genetic variation influences gene expression have revealed considerable discrepancies between genetic effects on mRNA versus protein abundance. Many gene expression QTLs alter protein abundance without detectable effects on mRNA levels (*Battle et al., 2015*; *Ghazalpour et al., 2011*; *Mirauta et al., 2020*; *Albert et al., 2014*; *Brion et al., 2020*; *Chick et al., 2016*; *Cenik et al., 2015*; *Abell et al., 2022*; *Foss et al., 2011*). We reasoned that protein-specific gene expression QTLs could arise through effects on UPS protein degradation. To test this idea and explore how variant effects on UPS activity influence other aspects of cellular physiology, we measured global gene expression at the protein and RNA levels in the same cultures of the BY strain and a BY strain engineered to contain the causal –469A>T RM allele in the *UBR1* promoter ("BY *UBR1* –469A>T"). As expected, the derived BY allele decreased *UBR1* protein and RNA levels (*Figure 5A/B*).

Out of 3,046 proteins quantified by mass spectrometry, 39 proteins were differentially abundant at a 10% FDR (*Figure 5A*, *Figure 5—source data 1*). Consistent with the reduced UPS activity conferred by the BY *UBR1* allele, a significant majority (28 of 39, 71%, binomial test $p$ = 9.5e-3) of differentially abundant proteins were increased by the BY allele (*Figure 5A*, *Figure 5—source data 1*). The median $\log_2$ fold change across all proteins was –0.012, while for differentially abundant proteins, the median $\log_2$ fold change was 0.37 (*Figure 5A*, *Figure 5—source data 1*). No Gene Ontology or Reactome pathway terms were enriched in our set of differentially abundant proteins. This result is consistent with recent observations that sequence features, rather than biological function or subcellular localization are the primary determinants of substrate targeting by E3 ligases. (*Kong et al., 2021*; *Christiano et al., 2020*).

To determine whether differences in protein abundance were reflected at the mRNA level, we used RNA-seq to quantify the levels of 5,675 transcripts. A total of 78 transcripts were differentially expressed between BY and BY *UBR1* –469A>T at a 10% FDR (*Figure 5B*, *Figure 5—source data 2*). Only three genes, *UBR1*, *HSP26*, and *TMA10*, showed significant and concordant changes at the RNA and protein levels (*Figure 5C*) and the overall correlation of $\log_2$ fold changes at the protein and mRNA levels was low, albeit significant (Pearson $r$ = 0.064, $p$ = 7.2e-4). In contrast to our proteomics results, the BY allele tended to decrease mRNA abundance, causing lower expression at a significant majority of differentially expressed genes (67 / 78, 86%, binomial test $p$ = 2e-6, *Figure 5B*, *Figure 5—source data 2*). The median $\log_2$ fold change across all transcripts was 0.0024, while at differentially expressed transcripts it was –0.20 (*Figure 5B*, *Figure 5—source data 2*). Multiple GO proteostasis-related pathways were enriched among the differentially abundant transcripts (*Figure 5—figure supplement 1*), driven by the decreased transcript abundance for genes such as *UBR1* and the chaperones *HSP26*, *HSP30*, *HSP31*, and *HSP82* in BY. Our results add to the emerging view of complex, protein-specific influences of genetic variation on gene expression (*Battle et al., 2015*; *Ghazalpour et al., 2011*; *Mirauta et al., 2020*; *Albert et al., 2014*; *Brion et al., 2020*; *Chick et al., 2016*; *Cenik et al., 2015*; *Abell et al., 2022*; *Foss et al., 2011*). Specifically, a non-coding variant that decreases expression of a single E3 ubiquitin ligase increases the levels of dozens of proteins without detectable effects on transcript abundance, implicating genetic effects on UPS activity as a potentially prominent source of post-translational variation in gene expression.

## Discussion

Protein degradation by the UPS is an essential biological process that influences virtually all aspects of eukaryotic cellular physiology (*Hanna and Finley, 2007*; *Varshavsky, 2011*; *Schwartz and Ciechanover, 1999*; *Finley and Prado, 2020*). Understanding the sources of variation in UPS activity thus has considerable implications for our understanding of numerous cellular and organismal traits, including human health and disease (*Schmidt and Finley, 2014*; *Petrucelli and Dawson, 2004*; *Gomes, 2013*; *Schwartz and Ciechanover, 1999*). Our statistically powerful, systematic genetic mapping of the N-end rule has revealed that individual genetic differences create heritable variation in UPS protein degradation. Genetic effects on UPS activity are numerous and comprise a continuous distribution of many loci with small effects and few loci of large effect (*Figure 2*), similar to other complex traits (*Mackay et al., 2009*; *Ehrenreich et al., 2009*). Previous efforts to understand how individual genetic differences cause variation in UPS activity have focused on individual disease-causing mutations in

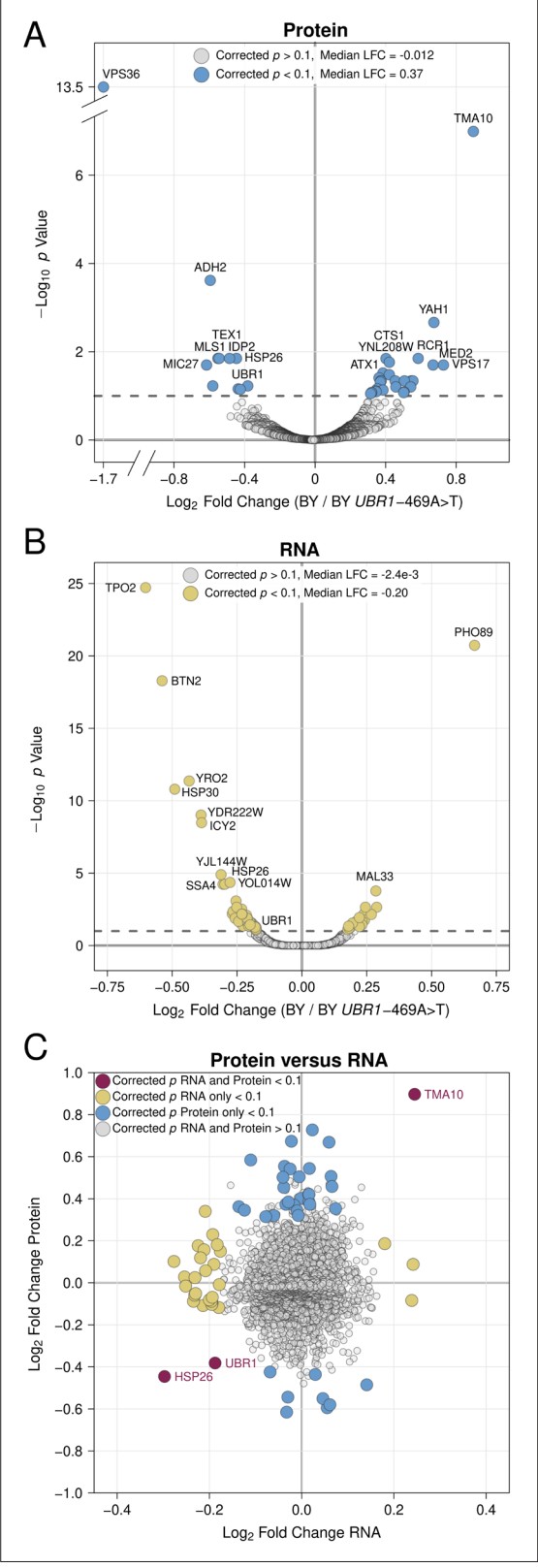

**Figure 5.** Proteomic and RNA-seq analysis of the effect of the *UBR1* –469A>T promoter variant on gene expression. (**A**) Protein fold-change versus statistical significance for BY versus BY *UBR1* –469A>T for all detected proteins. Differentially abundant proteins are shown in blue. (**B**) RNA fold-change versus statistical significance for BY versus BY *UBR1* –469A>T for all detected transcripts. Differentially expressed transcripts are shown in yellow.

*Figure 5 continued on next page*

*Figure 5 continued*

(**C**) Scatterplot comparing changes in protein and RNA abundance caused by *UBR1* –469A>T. "LFC" = log$_2$ fold change.

The online version of this article includes the following source data and figure supplement(s) for figure 5:

**Source data 1.** Full proteomics results.

**Source data 2.** Full RNA-seq results.

**Figure supplement 1.** Over-represented GO biological processes and Reactome pathways in the set of differentially expressed transcripts.

UPS genes (*Gomes, 2013*; *Agarwal et al., 2010*; *Zenker et al., 2006*; *Deng et al., 2011*; *Kröll-Hermi et al., 2020*). Our results show that these large-effect mutations in UPS genes sit atop one extreme of a continuous distribution of variant effects that is dominated by many loci of small effect. Aberrant UPS activity is a hallmark of many common diseases with a poorly-understood, complex genetic basis (*Schmidt and Finley, 2014*; *Petrucelli and Dawson, 2004*; *Zheng et al., 2014*). Our results raise the possibility that the effects of many common, small-effect alleles may contribute to the risk of these diseases through their effects on UPS activity.

Using genome engineering, we experimentally identified causal regulatory and missense variants in four functionally distinct ubiquitin system genes. A major function of the ubiquitin system is conferring specificity to UPS protein degradation (*Komander and Rape, 2012*; *Johnson et al., 1992*; *Bett, 2016*). Non-ubiquitinated proteins are blocked by the proteasome's 19S regulatory particle from degradation by the 20S catalytic core (*Inobe and Matouschek, 2014*). The selective binding of ubiquitinated substrates by the 19S regulatory particle ensures that only proteins targeted for degradation enter the proteasome. The activity of the ubiquitin system towards distinct substrates is highly variable, even for proteins degraded by the same UPS pathway (*Bachmair et al., 1986*; *Kats et al., 2018*; *Christiano et al., 2020*). Consistent with these observations, the effects of causal ubiquitin system gene variants were highly substrate-specific (*Figures 3 and 4*). Our results raise the question of whether UPS protein degradation is also shaped by variation in proteasome genes and whether any such effects would be less substrate-specific than those in the ubiquitin system. Given the multiple QTLs arising from ubiquitin system genes, detecting genetic influences on proteasome activity may benefit from assays that can measure proteasome activity independently of the ubiquitin system.

The remarkable complexity in causal variants we uncovered underscores the challenge of predicting variant effects on UPS protein degradation. Similar to recent results (*Lutz et al., 2022*; *Abell et al., 2022*), each of the four QTL regions we fine-mapped contained multiple causal variants in a single gene (*Figures 3 and 4*). In the case of *NTA1*, we observed that the effect of the D111E variant was likely masked during QTL mapping by the larger effect of the E129G variant (*Figure 4C*), highlighting the need to test individual variants in isolation. Causal variants may also exert substrate-specific effects on UPS protein degradation. We observed multiple instances where the magnitude of a causal variant's effect varied between substrates. In the case of *UBR1*, the RM *UBR1* ORF exerted significant, discrepant effects on the degradation of Arg/N-degrons (*Figure 3C*). Recent efforts have established that a protein's sequence critically determines how its degradation is altered by changes in UPS activity (*Christiano et al., 2020*). Thus, a complete understanding of a given variant's influence on UPS protein degradation will require testing its effect on the turnover of multiple substrates with diverse sequence compositions.

Our results suggest that genetic effects on UPS activity are an important source of post-translational variation in gene expression. A promoter variant that reduces UPS activity by decreasing *UBR1* expression alters the abundance of dozens of proteins without detectable effects on levels of the corresponding mRNA transcripts (*Figure 5*). Ubr1 and Doa10 target distinct sets of cellular proteins (*Kats et al., 2018*; *Kong et al., 2021*; *Christiano et al., 2020*). Their genes each contain multiple causal variants that differentially affected individual N-degrons and thus, potentially, endogenous cellular proteins. Similar effects arising from variation in the approximately 100 E3 ubiquitin ligases encoded in the *S. cerevisiae* genome (*Finley et al., 2012*) may help explain the numerous protein-specific gene expression QTLs (*Battle et al., 2015*; *Brion et al., 2020*; *Albert et al., 2014*; *Mirauta et al., 2020*). Such effects could be even more prevalent in the human genome, which encodes an estimated 600 E3 ubiquitin ligases (*Li et al., 2008*).

We have developed a generalizable framework for mapping genetic influences on protein degradation. Our results lay important groundwork for future efforts to understand how heritable differences in UPS activity contribute to variation in complex cellular and organismal traits, including the many diseases marked by aberrant UPS activity.

# Materials and methods

## Key resources table

| Reagent type (species) or resource | Designation | Source or reference | Identifiers | Additional information |
|---|---|---|---|---|
| Gene (*Saccharomyces cerevisiae*) | *UBR1* | Saccharomyces Genome Database (SGD) | YGR184C | edited to contain alternative alleles / variants |
| Gene (*S. cerevisiae*) | *DOA10* | SGD | YIL030C | edited to contain alternative alleles / variants |
| Gene (*S. cerevisiae*) | *UBC6* | SGD | YER100W | edited to contain alternative alleles / variants |
| Gene (*S. cerevisiae*) | *NTA1* | SGD | YJR062C | edited to contain alternative alleles / variants |
| Gene (*S. cerevisiae*) | *HIS3* | SGD | YOR202W | selectable marker for genome engineering |
| Gene (*S. cerevisiae*) | *LYP1* | SGD | YNL268W | selectable marker for genome engineering |
| Strain, strain background (*S. cerevisiae*) | BY4741 | Leonid Kruglyak | YFA0040 | *Supplementary file 5* |
| Strain, strain background (*S. cerevisiae*) | RM11.1a | Leonid Kruglyak | YFA0039 | *Supplementary file 5* |
| Strain, strain background (*S. cerevisiae*) | recombinant progeny of BY4741 x RM11.1a | this study | SFA- | *Supplementary file 5* |
| Strain, strain background (*S. cerevisiae*) | strains with tandem fluorescent timer reporters | this study | YFA- | *Supplementary file 5* |
| Strain, strain background (*S. cerevisiae*) | strains lacking individual ubiquitin-proteasome system genes | this study | YFA- | *Supplementary file 5* |
| Strain, strain background (*S. cerevisiae*) | strains with alternative UPS gene alleles / variants | this study | YFA- | *Supplementary file 5* |
| Strain, strain background (*Escherichia coli*) | DH5α | New England Biolabs | | for plasmid cloning and propagation |
| Recombinant DNA reagent | 23 plasmids | this study | PFA- | *Supplementary file 4* |
| Recombinant DNA reagent | backbone plasmid | Addgene | 35121 | |
| Recombinant DNA reagent | backbone plasmid | Addgene | 41030 | |
| Recombinant DNA reagent | KanMX cassette | *Wach et al., 1994*; 10.1002/yea.320101310 | | selectable marker for genome engineering |
| Recombinant DNA reagent | NatMX cassette | *Wach et al., 1994*; 10.1002/yea.320101310 | | selectable marker for genome engineering |
| Sequence-based reagent | 102 oligonucleotides | Integrated DNA Techologies | OFA- | *Supplementary file 3* |
| Commercial assay or kit | Nextera DNA Library Prep Kit | Illumina | FC-121–1030 | |
| Commercial assay or kit | EB Ultra II Directional RNA library kit for Illumina | New England Biolabs | E7760 | |
| Commercial assay or kit | Monarch Gel Extraction kit | New England Biolabs | T1010L | |
| Commercial assay or kit | HiFi DNA Assembly Cloning Kit | New England Biolabs | E5520S | |

*Continued on next page*

*Continued*

| Reagent type (species) or resource | Designation | Source or reference | Identifiers | Additional information |
|---|---|---|---|---|
| Commercial assay or kit | TMT10plex Isobaric Label Reagent Set | ThermoFisher Scientific | 90110 | |
| Commercial assay or kit | ZR Fungal / Bacterial RNA Miniprep kit | Zymo Research | R2014 | |
| Commercial assay or kit | Quick-96 DNA Plus kit | Zymo Research 10.1093/bioinformatics/btp324 | D4070 | |
| Software, algorithm | MULTIPOOL | *Edwards and Gifford, 2012*; 10.1186/1471-2105-13-S6-S8 | | |
| Software, algorithm | trimmomatic | *Bolger et al., 2014* 10.1093/bioinformatics/btu170 | | |
| Software, algorithm | kallisto | *Bray et al., 2016*; 10.1038/nbt.3519 | | |
| Software, algorithm | PANTHER | *Mi et al., 2021*; 10.1093/nar/gkaa1106 | | |
| Software, algorithm | fastp | *Chen et al., 2018*; 10.1093/bioinformatics/bty560 | | |
| Software, algorithm | RSeQC | *Wang et al., 2012*; 10.1093/bioinformatics/bts356 | | |
| Software, algorithm | Scaffold | https://www.proteomesoftware.com/ | | |
| Software, algorithm | Proteome Discoverer | Thermo Scientific | | |
| Software, algorithm | AlphaFold | *Jumper et al., 2021*; 10.1038/s41586-021-03819-2 | | |
| Software, algorithm | Inkscape | https://inkscape.org | | |
| Other | LSR II Flow Cytometer | BD | | flow cytometry |
| Other | FACSAria II Cell Sorter | BD | | cell sorting |
| Other | Orbitrap Fusion Tribrid MS-MS instrument | Thermo Scientific | | mass spectrometry |
| Other | Next-Seq 550 | Illumina | | DNA / RNAsequencing |

## Tandem fluorescent timer ubiquitin-proteasome system activity reporters

We used tandem fluorescent timers (TFTs) to measure ubiquitin-proteasome system (UPS) activity. TFTs are fusions of two fluorescent proteins (FPs) with distinct spectral profiles and maturation kinetics (*Khmelinskii et al., 2012*; *Khmelinskii and Knop, 2014*). In the most common implementation, a TFT consists of a faster maturing green fluorescent protein (GFP) and a slower maturing red fluorescent protein (RFP). Because the FPs in the TFT mature at different rates, the RFP / GFP ratio changes over time. If the degradation rate of a TFT exceeds the maturation rate of the RFP, the -$\log_2$ RFP / GFP ratio is directly proportional to the construct's degradation rate (*Khmelinskii and Knop, 2014*; *Khmelinskii et al., 2012*). When fused to N-degrons, the TFT's RFP / GFP ratio measures UPS N-end rule activity (*Khmelinskii et al., 2012*; *Khmelinskii et al., 2014*). The RFP / GFP ratio is also independent of the TFT's expression level, (*Khmelinskii et al., 2012*; *Khmelinskii and Knop, 2014*; *Kong et al., 2021*) preventing confounding from genetic effects on reporter expression in genetically diverse cell populations.

We used fluorescent proteins from previously characterized TFTs in our experiments (*Khmelinskii et al., 2016*; *Khmelinskii and Knop, 2014*; *Khmelinskii et al., 2012*; *Khmelinskii et al., 2014*). superfolder GFP (*Pédelacq et al., 2006*) (sfGFP) was used as the faster maturing FP in all TFTs. sfGFP matures in approximately 5 min and has excitation and emission maximums of 485 nm and 510 nm, respectively (*Pédelacq et al., 2006*). The slower maturing FP in each TFT was either mCherry or mRuby. mCherry matures in approximately 40 min and has excitation and emission maximums of 587 nm and 610 nm, respectively (*Shaner et al., 2004*). mRuby matures in approximately 170 min and has excitation and emission maximums of 558 nm and 605 nm, respectively (*Kredel et al., 2009*).

All TFT fluorescent proteins are monomeric. We separated green and red FPs in each TFT with an unstructured 35 amino acid linker sequence to minimize fluorescence resonance energy transfer (*Khmelinskii et al., 2012*).

## Construction of Arg/N-end and Ac/N-end pathway TFTs

To generate TFT constructs with defined N-terminal amino acids, we used the ubiquitin-fusion technique (*Bachmair et al., 1986*; *Varshavsky, 2005*; *Varshavsky, 2011*), which involves placing a ubiquitin moiety immediately upstream of a sequence encoding the desired N-degron. During translation, ubiquitin-hydrolases cleave the ubiquitin moiety, exposing the N-degron (*Figure 1A*). We synthesized DNA (Integrated DNA Technologies [IDT], Coralville, Iowa, USA) encoding the *Saccharomyces cerevisiae* ubiquitin sequence and a peptide linker sequence derived from *Escherichia coli* ß-galactosidase previously used to identify components of the Arg/N-end and Ac/N-end pathways (*Bachmair et al., 1986*). The peptide linker sequence is unstructured and contains internal lysine residues required for ubiquitination and degradation by the UPS (*Bachmair et al., 1986*; *Hwang et al., 2010*). Peptide linkers encoding the 20 possible N-terminal amino acids were made by PCR amplifying the linker sequence using oligonucleotides encoding each unique N-terminal amino acid (*Supplementary file 3*).

We then devised a general strategy to assemble TFT-containing plasmids with defined N-terminal amino acids (*Figure 1—figure supplement 2*). We first obtained sequences encoding each reporter element by PCR or DNA synthesis. We codon-optimized the sfGFP, mCherry, mRuby, and the TFT linker sequences for expression in *S. cerevisiae* using the Java Codon Adaptation Tool (JCaT) (*Grote et al., 2005*) and synthesized DNA fragments encoding each sequence (IDT). We used the *TDH3* promoter to drive expression of each TFT reporter. The *TDH3* promoter was PCR-amplified from Addgene plasmid #67639 (a gift from John Wyrick). We used the *ADH1* terminator in all TFT constructs, which we PCR amplified from Addgene plasmid #67639. We used the KanMX cassette (*Wach et al., 1994*), which confers resistance to G418, as the selection module for all TFT constructs and obtained this sequence by PCR amplification from Addgene plasmid #41030 (a gift from Michael Boddy). Thus, each construct has the general structure of *TDH3* promoter, N-degron, linker sequence, TFT, *ADH1* terminator, and the KanMX resistance cassette (*Figure 1—figure supplement 2*). Based on the half-lives of N-degrons (*Bachmair et al., 1986*; *Hwang et al., 2010*; *Varshavsky, 2011*), we used the mCherry-sfGFP TFT for all Arg/N-end constructs and the mRuby-sfGFP TFT for all Ac/N-end constructs.

We used Addgene plasmid #35121 (a gift from John McCusker) to construct all TFT plasmids. Digesting this plasmid with BamHI and EcoRV restriction enzymes produces a 2,451 bp fragment that we used as a vector backbone for TFT plasmid assembly. We obtained a DNA fragment containing 734 bp of sequence upstream of the *LYP1* start codon, a SwaI restriction site, and 380 bp of sequence downstream of the *LYP1* stop codon by DNA synthesis (IDT). We performed isothermal assembly cloning using the New England Biolabs (NEB; Ipswich, MA, USA) HiFi Assembly Cloning Kit (NEB) to insert the *LYP1* homology sequence into the BamHI/EcoRV digest of Addgene plasmid #35121 to create the final backbone plasmid BFA0190 (*Supplementary file 4*). We then combined SwaI digested BFA0190 and the components of each TFT reporter and used the NEB HiFi Assembly Kit (NEB) to produce each TFT plasmid. The 5′ and 3′ *LYP1* sequences in each TFT contain naturally-occurring SacI and BglII restriction sites, respectively. We digested each TFT plasmid with SacI and BglII (NEB) to obtain a linear DNA transformation fragment (*Figure 1—figure supplement 2*). The flanking *LYP1* homology and KanMX module in each TFT construct allows selection for reporter integration at the *LYP1* locus using G418 (*Goldstein and McCusker, 1999*) and the toxic amino acid analogue thialysine (S-(2-aminoethyl)-L-cysteine hydrochloride) (*Zwolshen and Bhattacharjee, 1981*; *Baryshnikova et al., 2010*; *Kuzmin et al., 2016*). The sequence identity of all assembled plasmids was verified by Sanger sequencing. The full list of plasmids used in this study is found in *Supplementary file 4*.

## Yeast strain handling

We used two strains of the yeast *Saccharomyces cerevisiae* to characterize our TFT reporters and perform genetic mapping of UPS activity. The haploid BY strain (genotype: *MATa his3Δ hoΔ*) is closely related to the *S. cerevisiae* S288C laboratory strain. The second mapping strain, RM, was originally isolated from a California vineyard and is haploid with genotype *MATa can1Δ::STE2pr-SpHIS5*

**Table 1.** Strain genotypes.

| Short Name | Genotype | Antibiotic Resistance | Auxotrophies |
|---|---|---|---|
| BY | *MATa his3Δ hoΔ* | | histidine |
| RM | *MATα can1Δ::STE2pr-SpHIS5* | clonNAT, hygromycin | histidine |
| | *his3Δ::NatMX hoΔ::HphMX* | | |
| BY *rpn4Δ* | *MATa his3Δ hoΔ rpn4Δ::NatMX* | clonNAT | histidine |
| BY *ubr1Δ* | *MATa his3Δ hoΔ ubr1Δ::NatMX* | clonNAT | histidine |
| BY *doa10Δ* | *MATa his3Δ hoΔ doa10Δ::NatMX* | clonNAT | histidine |

*his3Δ::NatMX AMN1-BY hoΔ::HphMX URA3-FY*. BY and RM differ at 1 nucleotide per 200 base pairs on average, such that approximately 45,000 single nucleotide variants (SNVs) between the strains can serve as markers in a genetic mapping experiment (*Albert et al., 2014*; *Brion et al., 2020*; *Ehrenreich et al., 2009*; *Ehrenreich et al., 2010*).

We built additional strains for characterizing our UPS activity reporters by deleting individual UPS genes from the BY strain. Each deletion strain was constructed by replacing the targeted gene with the NatMX cassette (*Goldstein and McCusker, 1999*), which confers resistance to the antibiotic nourseothricin. We PCR amplified the NatMX cassette from Addgene plasmid #35121 using primers with homology to the 5′ upstream and 3′ downstream sequences of the targeted gene. The oligonucleotides for each gene deletion cassette amplification are listed in *Supplementary file 3*. We created a BY strain lacking the *UBR1* gene, which encodes the Arg/N-end pathway E3 ligase Ubr1. We refer to this strain hereafter as 'BY *ubr1Δ*'. We created a BY strain ('BY *doa10Δ*') lacking the *DOA10* gene that encodes the Ac/N-end pathway E3 ligase Doa10. Finally, we created a BY strain ('BY *rpn4Δ*') lacking the *RPN4* that encodes the proteasome transcription factor Rpn4. *Table 1* lists these strains and their full genotypes. *Supplementary file 5* contains the complete list of strains used in this study.

*Table 2* describes the media formulations used for all experiments. Synthetic complete amino acid powders (SC -lys and SC -his -lys -ura) were obtained from Sunrise Science (Knoxville, TN, USA). Where indicated, we added the following reagents at the indicated concentrations to yeast media: G418, 200 mg/mL (Fisher Scientific, Pittsburgh, PA, USA); clonNAT (nourseothricin sulfate, Fisher Scientific), 50 mg/L; thialysine (S-(2-aminoethyl)-L-cysteine hydrochloride; MilliporeSigma, St. Louis, MO, USA), 50 mg/L; canavanine (L-canavanine sulfate, MilliporeSigma), 50 mg/L.

**Table 2.** Media formulations.

| Media Name | Abbreviation | Formulation |
|---|---|---|
| Yeast-Peptone-Dextrose | YPD | 10 g/L yeast extract |
| | | 20 g/L peptone |
| | | 20 g/L dextrose |
| Synthetic Complete | SC | 6.7 g/L yeast nitrogen base |
| | | 1.96 g/L amino acid mix -lys |
| | | 20 g/L dextrose |
| Haploid Selection | SGA | 6.7 g/L yeast nitrogen base |
| | | 1.74 g/L amino acid mix -his -lys -ura |
| | | 20 g/L dextrose |
| Sporulation | SPO | 1 g/L yeast extract |
| | | 10 g/L potassium acetate |
| | | 0.5 g/L dextrose |

## Yeast transformation

We used a standard yeast transformation protocol to construct reporter control strains and build strains with UPS activity reporters (*Gietz and Schiestl, 2007*). In brief, we inoculated yeast strains growing on solid YPD medium into 5 mL of YPD liquid medium for overnight growth at 30°C. The following morning, we diluted 1 mL of saturated culture into 50 mL of fresh YPD and grew the cells for 4 hr. The cells were then successively washed in sterile ultrapure water and transformation solution 1 (10 mM Tris HCl [pH 8.0], 1 mM EDTA [pH 8.0], and 0.1 M lithium acetate). At each step, we pelleted the cells by centrifugation at 3000 rpm for 2 min in a benchtop centrifuge and discarded the supernatant. The cells were suspended in 100 μL of transformation solution 1 along with 50 μg of salmon sperm carrier DNA and 300 ng of transforming DNA. The cells were incubated at 30 for 30 min and 700 μL of transformation solution 2 (10 mM Tris HCl [pH 8.0], 1 mM EDTA [pH 8.0], and 0.1 M lithium acetate in 40% polyethylene glycol [PEG]) was added to each tube, followed by a 30-min heat shock at 42°C. We then washed the transformed cells in sterile, ultrapure water. We added 1 mL of liquid YPD medium to each tube and incubated the tubes for 90 min with rolling at 30°C to allow for expression of the antibiotic resistance cassettes. After washing with sterile, ultrapure water, we plated 200 μL of cells on solid SC -lys medium with G418 and thialysine, and, for strains with the NatMX cassette, clonNAT. For each strain, we streaked multiple independent colonies (biological replicates) from the transformation plate for further analysis as indicated in the text. We verified reporter integration at the targeted genomic locus by colony PCR (*Ward, 1992*). The primers used for these experiments are listed in *Supplementary file 3*.

## Yeast mating and segregant populations

We created populations of genetically variable, recombinant cells ("segregants") for genetic mapping using a modified synthetic genetic array (SGA) approach (*Baryshnikova et al., 2010*; *Kuzmin et al., 2016*). We first mated BY strains with a given UPS activity reporter to RM by mixing freshly streaked cells of each strain on solid YPD medium. For each UPS activity reporter, we mated two independently-derived clones (biological replicates) to the RM strain. Cells were grown overnight at 30°C and we selected for diploid cells (successful BY-RM matings) by streaking mated cells onto solid YPD medium with G418 (which selects for the KanMX cassette in the TFT in the BY strain) and clonNAT (which selects for the NatMX cassette in the RM strain). We inoculated 5 mL of YPD with freshly streaked diploid cells for overnight growth at 30°C. The next day, we pelleted the cultures, washed them with sterile, ultrapure water, and resuspended the cells in 5 mL of SPO liquid medium (*Table 2*). We sporulated the cells by incubating them at room temperature with rolling for 9 days. After confirming sporulation by brightfield microscopy, we pelleted 2 mL of culture, washed cells with 1 mL of sterile, ultrapure water, and resuspended cells in 300 μL of 1 M sorbitol containing 3 U of Zymolyase lytic enzyme (United States Biological, Salem, MA, USA) to degrade ascal walls. Digestions were carried out at 30°C with rolling for 2 hr. We then washed the spores with 1 mL of 1 M sorbitol, vortexed for 1 min at the highest intensity setting, resuspended the cells in sterile ultrapure water, and confirmed the release of cells from ascii by brightfield microscopy. We plated 300 μl of cells onto solid SGA medium containing G418 and canavanine. This media formulation selects for haploid cells with (1) a UPS activity reporter via G418, (2) the *MATa* mating type via the *Schizosaccharomyces pombe HIS5* gene under the control of the *STE2* promoter (which is only active in *MATa* cells), and (3) replacement of the *CAN1* gene with *S. pombe HIS5* via the toxic arginine analog canavanine (*Baryshnikova et al., 2010*; *Kuzmin et al., 2016*). Haploid segregant populations were grown for 2 days at 30°C and harvested by adding 10 mL of sterile, ultrapure water and scraping the cells from each plate. We pelleted each cell suspension by centrifugation at 3000 rpm for 10 min and resuspended the cells in 1 mL of SGA medium. We added 450 μL of 40% (v/v) sterile glycerol solution to 750 μL of segregant culture and stored samples in screw cap cryovials at -80°C. We stored two independent sporulations of each reporter (derived from our initial matings) as independent biological replicates.

## Flow cytometry

We measured UPS activity by flow cytometry as follows. Yeast strains were manually inoculated into 400 μL of liquid SC -lys medium with G418 and grown overnight in 2 mL 96-well plates at 30°C with 1000 rpm mixing using a MixMate (Eppendorf, Hamburg, Germany). The following morning, we inoculated a fresh 400 μL of G418-containing SC -lys media with 4 μL of each saturated culture. Cells were

**Table 3.** Flow cytometry and FACS settings.

| Parameter | Laser Line (nm) | Laser Setting (V) | Filter |
|---|---|---|---|
| forward scatter (FSC) | 488 | 500 | 488/10 |
| side scatter (SSC) | 488 | 275 | 488/10 |
| sfGFP | 488 | 500 | 525/50 |
| mCherry | 561 | 615 | 610/20 |
| mRuby | 561 | 615 | 610/20 |

grown for an additional 3 hr prior to analysis by flow cytometry. All flow cytometry experiments were performed on an LSR II flow cytometer (BD, Franklin Lakes, NJ, USA) equipped with a 20 mW 488 nm laser with 488/10 and 525/50 filters for measuring forward/side scatter and sfGFP, respectively, as well as a 40 mW 561 nm laser and a 610/20 filter for measuring mCherry and mRuby. *Table 3* lists the parameters and settings that were used for all flow cytometry and fluorescence-activated cell sorting (FACS) experiments. We recorded 10,000 cells each from 8 independent biological replicates per strain for our analyses of BY, RM, and reporter control strains.

We analyzed flow cytometry data using R (R Foundation for Statistical Computing, Vienna Austria) and the flowCore R package (*Hahne et al., 2009*). We first filtered each flow cytometry dataset to include only those cells within 10% ± the forward scatter (a proxy for cell size) median. We empirically determined that this gating approach captured the central peak of cells in the FSC histogram. It also removed cellular debris, aggregates of multiple cells, and restricted our analyses to cells of the same approximate size. We observed that the TFT's output changed with the passage of time during flow cytometry experiments. We used the residuals of a loess regression of the TFT's output on time to correct for this effect, similar to a previously-described approach (*Brion et al., 2020*).

To characterize our TFT reporters, we used the following analysis steps. We extracted the median -$\log_2$ RFP / GFP ratio from each of 10,000 cells per strain per reporter. These values were Z-score normalized relative to the sample lowest degradation rate (typically the E3 ligase deletion strain). Following this transformation, the strain with lowest degradation rate has a degradation rate of approximately 0 and the now-scaled RFP / GFP ratio is directly proportional to the construct's degradation rate. To compare degradation rates between strains and individual UPS activity reporters, we then converted scaled RFP/GFP ratios to Z scores, which we report as "Normalized UPS Activity". Statistical significance was assessed using a one-way ANOVA with Tukey's HSD post-hoc test.

For fine-mapping causal genes and variants for UPS activity QTLs, we used the following approach. We extracted the median -$\log_2$ RFP / GFP ratio from each of 10,000 cells per strain per reporter. These values were Z-score normalized relative to the median of the control strain (a BY strain engineered to contain the BY allele of a candidate causal gene). Statistical significance was assessed using a t-test of each experimental strain versus the control strain with Benjamini-Hochberg correction for multiple testing (*Benjamini and Hochberg, 1995*).

## Fluorescence-activated cell sorting

We selected populations of segregants for QTL mapping using a previously described approach for isolating phenotypically extreme cell populations by FACS (*Albert et al., 2014*; *Brion et al., 2020*). Segregant populations were thawed approximately 16 hr prior to cell sorting and grown overnight in 5 mL of SGA medium containing G418 and canavanine. The following morning, 1 mL of cells from each segregant population was diluted into a fresh 4 mL of SGA medium containing G418 and canavanine. Segregant cultures were then grown for an additional 4 hours prior to sorting. All FACS experiments were carried out using a FACSAria II cell sorter (BD). We used plots of side scatter (SSC) height by SSC width and forward scatter (FSC) height by FSC width to remove doublets from each sample. We then filtered cells on the basis of FSC area, restricting our sorts to ±7.5% of the central FSC peak, which we empirically determined excluded cellular debris and aggregates while encompassing the primary haploid cell population. Finally, we defined a fluorescence-positive population by comparing each segregant population to negative control BY and RM strains without TFTs. We collected pools of 20,000 cells each from three gates drawn on each segregant population:

1. The 2% lower tail of the UPS activity distribution
2. The 2% upper tail of the UPS activity distribution
3. Fluorescence-positive cells without selection on UPS activity ("null pools"), which were used to determine the false positive rate of the QTL mapping method (see below)

We collected cell pools from two independent biological replicates (spore preparations) for each reporter. Each pool of 20,000 cells was collected into sterile 1.5 mL polypropylene tubes containing 1 mL of SGA medium and grown overnight at 30°C with rolling. The next day, we mixed 750 µL of cells with 450 µL of 40% (v/v) glycerol and stored this mixture in 2 mL 96-well plates at −80°C.

## Genomic DNA isolation and library preparation

We extracted genomic DNA from sorted segregant pools for whole-genome sequencing. Deep-well plates containing glycerol stocks of sorted segregant pools were thawed and 800 µL of each sample was pelleted by centrifugation at 3,700 rpm for 10 min. We discarded the supernatant and resuspended cell pellets in 800 µL of a 1 M sorbitol solution containing 0.1 M EDTA, 14.3 mM β-mercaptoethanol, and 500 U of Zymolyase lytic enzyme to digest cell walls prior to DNA extraction. The digestion reaction was carried out by resuspending cell pellets with mixing at 1,000 rpm for 2 min followed by incubation for 2 hr at 37°C. When the digestion reaction finished, we discarded the supernatant, resuspended cells in 50 µL of phosphate buffered saline, and used the Quick-DNA 96 Plus kit (Zymo Research, Irvine, CA, USA) to extract genomic DNA. We followed the manufacturer's protocol to extract genomic DNA with the following modifications. We incubated cells in a 20 mg/mL proteinase K solution overnight with incubation at 55°C. After completing the DNA extraction protocol, we eluted DNA using 40 µL of DNA elution buffer (10 mM Tris-HCl [pH 8.5], 0.1 mM EDTA). The DNA concentration for each sample was determined using the Qubit dsDNA BR assay kit (Thermo Fisher Scientific, Waltham, MA, USA) in a 96 well format using a Synergy H1 plate reader (BioTek Instruments, Winooski, VT, USA).

We used a previously-described approach to prepare libraries for short-read whole-genome sequencing on the Illumina Next-Seq platform (*Albert et al., 2014*; *Brion et al., 2020*). We used the Nextera DNA library kit (Illumina, San Diego, CA, USA) according to the manufacturer's instructions with the following modifications. For the tagmentation reaction, 5 ng of genomic DNA from each sample was diluted in a master mix containing 4 µL of Tagment DNA buffer, 1 µL of sterile molecular biology grade water, and 5 µL of Tagment DNA enzyme diluted 1:20 in Tagment DNA buffer. The tagmentation reaction was run on a SimpliAmp thermal cycler (Thermo Fisher Scientific) using the following parameters: 55°C temperature, 20 µL reaction volume, 10 min incubation. To prepare libraries for sequencing, we added 10 µL of the tagmentation reaction to a master mix containing 1 µL of an Illumina i5 and i7 index primer pair mixture, 0.375 µL of ExTaq polymerase (Takara Bio, Mountain View, CA, USA), 5 µL of ExTaq buffer, 4 µL of a dNTP mixture, and 29.625 µL of sterile molecular biology grade water. We generated all 96 possible index oligo combinations using 8 i5 and 12 i7 index primers. The library amplification reaction was run on a SimpliAmp thermal cycler with the following parameters: initial denaturation at 95°C for 30 s, then 17 cycles of 95°C for 10 s (denaturation), 62°C for 30 s (annealing), and 72°C for 3 min (extension). We quantified the DNA concentration of each reaction using the Qubit dsDNA BR assay kit (Thermo Fisher Scientific) and pooled 10 µL of each reaction. This pooled mixture was run on a 2% agarose gel and we extracted and purified DNA in the 400 bp to 600 bp region using the Monarch Gel Extraction Kit (NEB) according to the manufacturer's instructions.

## Whole-genome sequencing

We submitted pooled, purified DNA libraries to the University of Minnesota Genomics Center (UMGC) for Illumina sequencing. Prior to sequencing, UMGC staff performed three quality control (QC) assays. Library concentration was determined using the PicoGreen dsDNA quantification reagent (Thermo Fisher Scientific) with libraries at a concentration of 1 ng/µL passing QC. Library size was determined using the Tapestation electrophoresis system (Agilent Technologies, Santa Clara, CA, USA) with libraries in the range of 200–700 bp passing QC. Library functionality was determined using the KAPA DNA Library Quantification kit (Roche, Penzberg, Germany), with libraries with a concentration greater than 2 nM passing. All submitted libraries passed each QC assay. We submitted 7 libraries for sequencing at different times. Libraries were sequenced on a NextSeq 550 instrument

(Illumina). Depending on the number of samples, we used the following output settings. For libraries with 70 or more samples (2 libraries), 75 bp paired end sequencing was performed in high-output mode to generate approximately $360 \times 10^6$ reads. For libraries with 50 or fewer samples (5 libraries), 75 bp paired end sequencing was performed in mid-output mode to generate approximately $120 \times 10^6$ reads. Average read coverage of the genome ranged from 9 to 35 with a median coverage of 28 across all libraries. Sequence data de-multiplexing was performed by UMGC. Whole-genome sequencing data have been deposited into the NIH Sequence Read Archive under Bioproject accession PRJNA881749.

## Raw whole-genome sequencing data processing

We calculated allele frequencies from our whole-genome sequencing data using the following pipeline. We initially filtered reads to include only those reads with mapping quality scores greater than 30. We aligned the filtered reads to the *S. cerevisiae* reference genome (version sacCer3) using BWA (*Li and Durbin, 2009a*) (command: 'mem -t 24'). We then used samtools (*Li et al., 2009b*) to remove unaligned reads, non-uniquely aligned reads, and PCR duplicates (command: 'samtools rmdup -S'). Finally, we produced vcf files containing coverage and allelic read counts at each of 18,871 high-confidence, reliable SNPs (*Bloom et al., 2013*; *Ehrenreich et al., 2010*) (command: 'samtools mpileup -vu -t INFO/AD -l'). Because the BY strain is closely related to the S288C genome reference *S. cerevisiae* strain, we considered BY alleles reference and RM alleles alternative alleles.

## QTL mapping

We identified QTLs from sequence data following established procedures for bulk segregant analysis (*Ehrenreich et al., 2010*; *Albert et al., 2014*; *Brion et al., 2020*). Allele counts in the vcf files generated above were provided to the MULTIPOOL algorithm (*Edwards and Gifford, 2012*). MULTIPOOL computes logarithm of the odds (LOD) scores by comparing two models: (1) a model in which the high and low UPS activity pools come from one from common population and thus share the same frequency of the BY and RM allele, and (2) a model in which these pools come from two populations with two different allele frequencies, indicating the presence of a QTL. We identified QTLs as genomic regions exceeding an empirically-derived significance threshold (see below). We used MULTIPOOL with the following settings: bp per centiMorgan = 2,200, bin size = 100 bp, effective pool size = 1,000. As in previous QTL mapping in the BY/RM cross by bulk segregant analysis (*Albert et al., 2014*; *Brion et al., 2020*), we excluded variants with allele frequencies higher than 0.9 or lower than 0.1 (*Albert et al., 2014*; *Brion et al., 2020*). We also used MULTIPOOL to estimate confidence intervals for each significant QTL, which we defined as a 2-LOD drop from the QTL peak position. To visualize QTLs and gauge their effects, we also computed the RM allele frequency differences (ΔAF) at each site between our high and low UPS activity pools. Because allele frequencies are affected by random counting noise, we used loess regression to smooth the allele frequency for each sample before computing ΔAF. We used the smoothed values to plot the ΔAF distribution along the genome and as a measure of QTL effect size.

## Null sorts and empirical false discovery rate estimation

We used "null" segregant pools (fluorescence-positive cells with no selection on UPS activity) to empirically estimate the false discovery rate (FDR) of our QTL mapping method. Because these cells are obtained as two pools from the same null population in the same sample, any ΔAF differences between them are the result of technical noise or random variation. We permuted these null comparisons across segregant pools with the same UPS activity reporter for a total of 112 null comparisons. We define the "null QTL rate" at a given LOD threshold as the number of QTLs that exceeded the threshold in these comparisons divided by the number of null comparisons. To determine the FDR for a given LOD score, we then determined the number of QTLs for our experimental comparisons (high UPS activity versus low UPS activity). We define the "experimental QTL rate" as the number of experimental QTLs divided by the number of experimental comparisons. The FDR is thus computed as follows:

$$null\ QTL\ rate = \frac{n.\ null\ QTLs}{n.\ null\ comparisons}$$

$$experimental\ QTL\ rate = \frac{n.\ experimental\ QTLs}{n.\ experimental\ comparisons}$$

$$FDR = \frac{null\ QTL\ rate}{experimental\ QTL\ rate}$$

We evaluated the FDR over a LOD range of 2.5–10 in 0.5 LOD increments. We found that a LOD value of 4.5 led to a null QTL rate of 0.0625 and an FDR of 0.507%. We used this value as our significance threshold for QTL mapping and further filtered our QTL list by excluding QTLs that were not detected in each of two independent biological replicates. Replicating QTLs were defined as those whose peaks were within 100 kb of each other on the same chromosome with the same direction (positive or negative) of RM allele frequency difference between high and low UPS activity pools.

## QTL fine-mapping by allelic engineering

We used 'CRISPR-Swap' (*Lutz et al., 2019*), a two-step method for scarless allelic editing, to fine-map QTLs to the level of their causal genes and nucleotides. In the first step of CRISPR-Swap, a gene of interest (GOI) is deleted and replaced with a selectable marker. In the second step, cells are co-transformed with (1) a plasmid that expresses CRISPR-cas9 and a guide RNA targeting the selectable marker and (2) a repair template encoding the desired allele of the GOI.

We used CRISPR-Swap to generate BY strains harboring either RM alleles or chimeric BY/RM alleles of several genes, as described below. To do so, we first replaced the gene of interest in BY with the NatMX selectable marker by transforming a PCR product encoding the NatMX cassette with 40 bp overhangs at the 5' and 3' ends of the targeted gene. To generate *GOIΔ::NatMX* transformation fragments, we PCR amplified NatMX from Addgene plasmid #35121 with the primers listed in ***Supplementary file 3*** using Phusion Hot Start Flex DNA polymerase (NEB). The NatMX cassette was transformed into the BY strain using the methods described above and transformants were plated onto YPD medium containing clonNAT. We verified the deletion of each gene of interest from single-colony purified transformants by colony PCR (primer sequences listed in ***Supplementary file 3***).

We then modified the original CRISPR-Swap plasmid (PFA0055, Addgene plasmid #131774) to replace its *LEU2* selectable marker with the *HIS3* selectable marker, creating plasmid PFA0227 (***Supplementary file 4***). To build PFA0277, we first digested PFA0055 with restriction enzymes BsmBI-v2 and HpaI to remove the *LEU2* selectable marker. We synthesized the *S. cerevisiae HIS3* selectable marker from plasmid pRS313 (***Sikorski and Hieter, 1989***) with 20 base pairs of overlap to BsmBI-v2/HpaI-digested PFA0055 on both ends. We used this synthetic *HIS3* fragment and BsmBI-v2/HpaI-digested PFA0055 to create plasmid PFA0227 by isothermal assembly cloning using the HiFi Assembly Cloning Kit (NEB) according to the manufacturer's instructions. In addition to the *HIS3* selectable marker, PFA0227 contains the cas9 gene driven by the constitutively active *TDH3* promoter and a guide RNA, gCASS5a, that directs cleavage of a site immediately upstream of the *TEF* promoter used to drive expression of the MX series of selectable markers (***Goldstein and McCusker, 1999***; ***Lutz et al., 2019***). We verified the sequence of PFA0227 by Sanger sequencing.

We used genomic DNA from BY and RM strains as a template to PCR amplify repair templates for CRISPR-Swap. Genomic DNA was extracted from BY and RM strains using the '10-min prep' protocol (***Hoffman and Winston, 1987***). We amplified full-length repair templates from RM and BY containing each GOI's promoter, open-reading frame (ORF), and terminator using Phusion Hot Start Flex DNA polymerase (NEB). We also created chimeric repair templates containing combinations of BY and RM alleles using PCR splicing by overlap extension (***Horton et al., 1989***). ***Table 4*** lists the repair templates used for CRISPR swap. The sequence of all repair templates was verified by Sanger sequencing.

To create allele swap strains, we co-transformed BY strains with 200 ng of plasmid PFA0227 and 1.5 µg of GOI repair template. Transformants were selected and single colony purified on synthetic complete medium lacking histidine and then patched onto solid YPD medium. We tested each strain for the desired exchange of the NatMX selectable marker with a *UBR1* allele by patching strains onto solid YPD medium containing clonNAT. We then verified allelic exchange in strains lacking clonNAT resistance by colony PCR (primers listed in ***Supplementary file 3***). We kept 16 independently-derived biological replicates of each allele swap strain. To test the effects of each allele swap, we transformed UPS activity reporters into our allele swap strains and characterized reporter activity by flow cytometry using the methods described above.

**Table 4.** CRISPR-swap repair templates.

| Gene | Allele Name | Promoter | ORF | Terminator |
|------|-------------|----------|-----|------------|
| *UBR1* | *UBR1* BY | BY | BY | BY |
| *UBR1* | *UBR1* RM | RM | RM | RM |
| *UBR1* | *UBR1* RM promoter | RM | BY | BY |
| *UBR1* | *UBR1* RM ORF | BY | RM | BY |
| *UBR1* | *UBR1* RM terminator | BY | BY | RM |
| *UBR1* | *UBR1* -469A>T | –469, RM; all other, BY | BY | BY |
| *UBR1* | *UBR1* -197T>G | –197, RM; all other, BY | BY | BY |
| *DOA10* | *DOA10* BY | BY | BY | BY |
| *DOA10* | *DOA10* RM | RM | RM | RM |
| *DOA10* | *DOA10* Q410E | BY | 1228, RM; all other, BY | BY |
| *DOA10* | *DOA10* K1012N | BY | 3036, RM; all other, BY | BY |
| *DOA10* | *DOA10* Y1186F | BY | 3557, RM; all other, BY | BY |
| *NTA1* | *NTA1* BY | BY | BY | BY |
| *NTA1* | *NTA1* RM | RM | RM | RM |
| *NTA1* | *NTA1* RM promoter | RM | BY | BY |
| *NTA1* | *NTA1* D111E | RM | 331, RM; all other, BY | BY |
| *NTA1* | *NTA1* E129G | RM | 386, RM; all other, BY | BY |
| *UBC6* | *UBC6* BY | BY | BY | BY |
| *UBC6* | *UBC6* RM | RM | RM | RM |
| *UBC6* | *UBC6* RM promoter | RM | BY | BY |
| *UBC6* | *UBC6* D229G | BY | 1686, RM; all other, BY | BY |
| *UBC6* | *UBC6* RM terminator | BY | BY | RM |

We tested whether a QTL on chromosome V results from variation in *UBC6* using CRISPR-Swap. Deleting *UBC6* caused a large growth defect relative to the wild-type BY strain. Providing cells with multiple *UBC6* alleles, including the BY allele, did not correct the growth rate defect. We did not observe growth defects in any other fine-mapping strains.

## RNA isolation

We isolated total RNA from 5 independent biological replicates each of the wild-type BY strain and a BY strain edited to contain the –469A>T RM variant in the *UBR1* promoter (hereafter "*UBR1* –469A>T BY"). All 10 samples were grown and harvested at the same time. BY and *UBR1* –469A>T BY strains were grown overnight at 30°C in 5 mL of SC medium. The following day, the cultures were diluted to an OD of 0.05 in 100 mL of fresh SC medium and grown for approximately 7 hours. When the optical density (OD) of each culture was approximately 0.40, the cells were pelleted by centrifugation at 3,000 rpm for 10 min. Pellets were then washed by resuspending them in 1 mL of sterile ultrapure water, followed by centrifugation at 3,000 rpm for 3 min to again pellet the cells. Following this step, cell pellets were resuspended in 1 mL of ultrapure water and split into 4 aliquots, each containing 250 μL. After re-centrifuging and discarding the supernatant, the pellets were snap frozen by immersion in liquid nitrogen, followed by storage at –80°C. Pellets were subsequently used for RNA isolation and mass spectrometric proteomic analysis, as described below.

Total RNA was extracted from frozen cell pellets using the ZR Fungal/Bacterial miniprep kit (Zymo), according to the manufacturer's instructions. Briefly, total RNA was isolated from cell pellets in two batches, each containing equal numbers of BY and *UBR1* –469A>T BY samples. After thawing, pellets were resuspended in lysis buffer and transferred to screwcap lysis tubes containing glass beads. Tubes

were secured in a Mini-BeadBeater (BioSpec Products, Bartlesville, OK, USA) and cells were processed in 5 cycles of 2 min of agitation followed by 2 min at −80°C. The cell lysate/bead mixture was centrifuged for 1 min at 16,000 x g and 400 µL of 95% ethanol was added to the cleared supernatant followed by mixing. Samples were then spun through a binding column and on-column DNA digest was performed with DNase I (Zymo) according to the manufacturer's instructions. Total RNA was eluted from columns using 50 µL of RNase-free ultrapure water. The concentration of each sample was quantified using RiboGreen; all samples had a concentration greater than 300 ng/µL. The integrity of each sample was assessed at UMGC using the Tapestation (Agilent) and an RNA ScreenTape. RNA integrity numbers ranged from 9.7 to 10.0 (where 10.0 is the maximum possible score), with a median value of 9.9. All RNA samples were stored at −80°C.

## RNA-seq

We isolated mRNA from each total RNA sample using the 550 ng of total RNA input and the NEBNext Poly(A) mRNA Magnetic Isolation Module (NEB). All samples were processed in a single batch and the isolated mRNA from each sample was used to prepare RNA sequencing libraries using the NEBNext Ultra II Directional RNA Library Prep kit (NEB) according to the manufacturer's instructions. Libraries were amplified using NEBNext Ultra II Q5 polymerase and unique combinations of primers from the NEBNext Multiplex Oligos for Illumina (NEB). The following amplification protocol was used: initial denaturation at 98°C for 30 s, followed by 10 cycles of 98°C (10 s; denaturation), 65°C (75 s; annealing and extension), and a 65°C final extension for 5 min. PCR reactions were pooled using equal amounts of DNA and submitted to UMGC for three quality control assays, which measured the library concentration by PicoGreen, library functionality by KAPA qPCR, and library size using the Tapestation electrophoresis system (Agilent). The resulting library contained a small amount of adapter dimer (approximately 9%), which was subsequently removed via a bead-based cleanup. The final, cleaned library passed all three QC assays and was sequenced on a Next-Seq 2000 instrument (Illumina) in paired-end mode with 150 bp reads. The sequencing run generated 1,367,252,076 reads with an average of 136,725,207 (range: 112,285,619–152,571,763) reads per sample.

## RNA-seq data processing and analysis

We performed quality control and preprocessing of RNA-seq data using fastp (*Chen et al., 2018*). Our initial processing removed reads with a length less than 36 bp and any reads where the mean quality dropped below a mean quality score of 15 in a 4 bp window. We also used fastp to trim adapter sequences from the ends of all reads. We then used Kallisto (*Bray et al., 2016*) to pseudo-align processed reads to the *S. cerevisiae* transcriptome, which was obtained from Ensembl (version 96) (*Howe et al., 2021*).

To identify differentially expressed transcripts, we used the estimated counts obtained from Kallisto as a measure of gene expression and filtered the estimated counts using the following procedures. First, we computed a transcript Transcript Integrity Number (TIN) for each gene using the RSeqQC (*Wang et al., 2012*) and removed any genes with a TIN less than 1 for any sample. We also removed any genes that Kallisto estimated to have an effective length less of less than 1 and those genes whose estimated counts were less than 10 in any sample. The resulting dataset comprised 5,676 expressed genes. Raw RNA-seq reads and processed counts were deposited in the NIH Gene Expression Omnibus database under accession number GSE213689. We used DESeq2 (*Love et al., 2014*) to perform statistical analysis of the resulting dataset. We used the RNA harvest batch and OD at time of sample harvest as covariates in our analysis. To further control for confounding sample-to-sample variation, we used surrogate variable analysis (*Leek et al., 2012*; *Leek and Storey, 2007*), which identified two significant surrogate variables that were subsequently added to our statistical model. We corrected for multiple testing using the Benjamini-Hochberg method (*Benjamini and Hochberg, 1995*) and considered significant differences as those with a corrected *p*-value less than 0.1.

To link differences in transcript abundance to biological pathways, we performed gene ontology enrichment analysis using PANTHER (*Mi et al., 2021*). The 'statistical overrepresentation test' was used to search for gene ontology (GO) biological processes and Reactome pathways enriched in our set of 78 transcripts differentially expressed between BY and *UBR1* –469A>T BY. We used the 5,676 genes quantified in our RNA-seq statistical analysis as the reference set and used the Benjamini-Hochberg

method (*Benjamini and Hochberg, 1995*) to correct for multiple testing. GO terms and Reactome pathways with a corrected *p*-value less than 0.05 were considered significant in our analysis.

## Protein isolation and proteomic analysis by mass spectrometry

To quantify gene expression at the protein level, we submitted five cell pellets each from the same BY and *UBR1* –469A>T BY cultures used for RNA-seq analysis to the University of Minnesota Center for Mass Spectrometry and Proteomics (CMSP) for proteomic analysis by mass spectrometry. Cell pellets were resuspended for protein extraction in a protein extraction buffer containing 7 M urea, 2 M thiourea, 0.4 M triethylammonium bicarbonate pH 8.5, 20% acetonitrile, and 4 mM tris(2-carboxyethyl)phosphine. Cell lysis and protein extraction was then performed using the Barocycler NEP2320 (Pressure Biosciences, Medford, MA, USA).

Samples were prepared and analyzed by mass spectrometry as follows. CMSP first labeled individual samples using the tandem mass tag (TMT) 10plex labeling kit (Thermo). After tagging, samples were pooled for analysis by mass spectrometry on an Orbitrap Tribrid Eclipse instrument (Thermo). Database searching was performed using the Proteome Discoverer software and the statistical analysis of protein abundance was performed in Scaffold (Proteome Software, Portland, OR, USA). We considered proteins to be differentially abundant between strains if they had a and a Benjamini-Hochberg corrected *p*-value less than 0.1. We performed ontological enrichment analysis of differentially abundant proteins using PANTHER as described above, except that the set of 3,046 detected proteins was used as the reference set.

## Evolutionary analysis of variants

We inferred the allelic status of individual variants by comparing them to two outgroups: a likely-ancestral Taiwanese *S. cerevisiae* isolate and the sister species *Saccharomyces paradoxus*. We classified variants as ancestral if they were found in at least one outgroup. All alleles analyzed in this study could be unambiguously classified using this approach. We extracted the population frequency of all analyzed variants using genome sequence data from a panel of 1,011 *S. cerevisiae* isolates (*Peter et al., 2018*).

## Data and statistical analysis

All data were analyzed using R (version 3.6.1; R Project for Statistical Computing). For all boxplots, the center line shows the median, the box excludes the upper and lower quartiles, the whiskers extend to 1.5 times the interquartile range. Protein structure predictions were obtained from the AlphaFold Protein Structure Database (*Jumper et al., 2021*) and visualized using ChimeraX (*Pettersen et al., 2021*). DNA binding motifs were determined using the Yeast Transcription Factor Specificity Compendium database (*de Boer and Hughes, 2012*). Final figures and illustrations were made using Inkscape (version 0.92; Inkscape Project).

Computational scripts used to process data, for statistical analysis, and to generate figures are available at: https://www.github.com/mac230/N-end_Rule_QTL_paper; copy archived at swh:1:rev:24baa12af4e9c45691be2590ab30b2c1faf0c497 (*Collins, 2022*).

## Acknowledgements

We thank Leonid Kruglyak for the BY and RM yeast strains and Michael Knop for technical assistance in implementing the TFT reporter system. We thank the University of Minnesota's Flow Cytometry Resource, Genomics Center, and Center for Mass Spectrometry and Proteomics for their contributions to the project. We thank Margaret Kliebhan for the BY / RM variant file used for QTL mapping. We thank the members of the Albert laboratory and the BioKansas Scientific Writing Program for critical feedback on the manuscript. This work was supported by NIH grants F32-GM128302 to MAC and R35-GM124676 to FWA, as well as a Pew Scholarship in the Biomedical Sciences from the Pew Charitable Trusts to FWA.

## Additional information

### Funding

| Funder | Grant reference number | Author |
|---|---|---|
| National Institutes of Health | F32-GM128302 | Mahlon A Collins |
| National Institutes of Health | R35-GM124676 | Frank Wolfgang Albert |
| Pew Charitable Trusts | Scholarship in the Biomedical Sciences | Frank Wolfgang Albert |

The funders had no role in study design, data collection and interpretation, or the decision to submit the work for publication.

### Author contributions

Mahlon A Collins, Conceptualization, Software, Formal analysis, Supervision, Funding acquisition, Validation, Investigation, Visualization, Methodology, Writing - original draft, Project administration, Writing – review and editing; Gemechu Mekonnen, Formal analysis, Validation, Investigation; Frank Wolfgang Albert, Conceptualization, Resources, Supervision, Funding acquisition, Methodology, Writing – review and editing

### Author ORCIDs

Mahlon A Collins ⓘ http://orcid.org/0000-0001-6799-5645
Frank Wolfgang Albert ⓘ http://orcid.org/0000-0002-1380-8063

### Decision letter and Author response

Decision letter https://doi.org/10.7554/eLife.79570.sa1
Author response https://doi.org/10.7554/eLife.79570.sa2

## Additional files

### Supplementary files

• Supplementary file 1. Allele frequency difference and LOD score traces from QTL mapping experiments. The plots show the loess-smoothed RM allele frequency difference (high UPS activity pool minus low UPS activity pool) and LOD score traces for the 20 N-degrons. QTLs are marked with asterisks, which are colored by biological replicate.

• Supplementary file 2. Influence of LOD score significance threshold on QTL pathway specificity. The LOD score and RM allele frequency difference (QTL effect direction) traces for two independent biological replicates of each N-degron are shown for each of 23 pathway-specific QTL regions. Dashed lines at distinct LOD scores illustrate how changing the significance threshold changes the pathway-specificity of a given QTL region.

• Supplementary file 3. Oligonucleotides. Table listing oligonucleotides used in this study.

• Supplementary file 4. Plasmids. Table of plasmids used in this study.

• Supplementary file 5. Yeast strains. Table listing all yeast strains used in the study.

• MDAR checklist

### Data availability

Raw sequencing reads from QTL mapping experiments are available from the NIH Sequence Read Archive under the Bioproject Accession PRJNA881749. Raw and processed RNA-seq data is available from the NIH Gene Expression Omnibus under the accession GSE213689. These datasets are fully available without restriction. Computational scripts used to process data, for statistical analysis, and to generate figures are available at: https://www.github.com/mac230/N-end_Rule_QTL_paper, (copy archived at swh:1:rev:24baa12af4e9c45691be2590ab30b2c1faf0c497).

The following datasets were generated:

| Author(s) | Year | Dataset title | Dataset URL | Database and Identifier |
|---|---|---|---|---|
| Collins MA | 2022 | Variation in ubiquitin system genes creates substrate-specific effects on proteasomal protein degradation | http://www.ncbi.nlm.nih.gov/geo/query/acc.cgi?acc=GSE213689 | NCBI Gene Expression Omnibus, GSE213689 |
| Collins MA | 2022 | Variation in ubiquitin system genes creates substrate-specific effects on proteasomal protein degradation | https://www.ncbi.nlm.nih.gov/bioproject?term=PRJNA881749 | NCBI BioProject, PRJNA881749 |

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

# Appendix 1

## Genetic Influences on the Proline N-degron

We observed that, consistent with previous results (*Hwang et al., 2010*; *Gilchrist et al., 1997*), the proline N-degron TFT was only partially stabilized in BY *DOA10Δ* (*Figure 1D*, *Figure 1—figure supplement 1*, *Figure 1—source data 1*). Ubiquitin is inefficiently cleaved in the ubiquitin-fusion technique when followed by a proline N-degron (*Varshavsky, 2011*; *Varshavsky, 2005*), leading to the production of two species, proline N-degron constructs and constructs with uncleaved N-terminal ubiquitin moieties. N-terminal ubiquitin functions as a degron and is recognized and degraded by the ubiquitin-fusion degradation (UFD) UPS pathway (*Johnson et al., 1995*). The proline N-end TFT thus measures the activities of the Ac/N-end pathway towards the proline N-degron and the UFD pathway towards the N-terminal ubiquitin fusion degron.

Despite this partial limitation, we were able to map genetic influences on the proline N-degron. We detected 5 Ac/N-end pathway-specific QTLs using our proline reporter, including those resulting from variation in *DOA10* and *UBC6*. There were no additional QTLs affecting the majority of Ac/N-end reporters that were not detected with the proline reporter (*Figure 2B*, *Figure 2—source data 2*). We therefore conclude that the QTLs identified with the proline reporter correspond primarily to true genetic influences on the proline N-degron.

Three QTLs, on one chromosome XII and two on chromosome XVI were detected only with the proline reporter (*Figure 2B*, *Figure 2—source data 2*). The intervals for these QTLs did not contain genes previously linked to either the proline N-degron, the Ac/N-end pathway, or the UFD pathway.

