## [Editor Report]

The authors use an elegant experimental design to study genetic variation in the ubiquitin-proteasome degradation system in yeast. They identify a large number of QTLs for naturally occurring variation, and they elucidate the causal variants and likely functional mechanisms of several of these. The paper illustrates an innovative new approach to high-throughput QTL mapping for specific molecular processes.

---

## [Decision Letter]

**Decision letter after peer review:**

Thank you for submitting your article "Variation in Ubiquitin System Genes Creates Substrate-Specific Effects on Proteasomal Protein Degradation" for consideration by *eLife*. Your article has been reviewed by 4 peer reviewers, , including Magnus Nordborg as the Reviewing Editor and Reviewer #1 and the evaluation has been overseen by David Ron as the Senior Editor.

The reviewers have discussed their reviews with one another and agreed this work is elegant, and absolutely deserves to be published. The Reviewing Editor has drafted this to help you prepare a revised submission – please also see the individual reviews for further suggestions for improvements.

There is an overall need to quantify several statements (see individual reviews). This should be straightforward and requires no additional experiments.

In addition, we would like to encourage you to put a bit more thought into the evolutionary analysis. For example, the issue of pleiotropy could be discussed further (as noted by Reviewer 1), and so could the persistent shifts in effect between the strains (Reviewer 4). It may be worth considering Hunter Fraser's use of sign tests to detect selection, e.g.,

https://www.pnas.org/doi/10.1073/pnas.0912245107 in this context.

*Reviewer #1 (Recommendations for the authors):*

I trust more mechanistic reviewers to comment on your experiments. From my point of view, I have only a few suggestions for improvements:

1) You argue (in connection with Figure 2)1 that "many QTLs were found only for individual pathways", but this is surely sensitive to significance thresholds. A more sophisticated analysis of pleiotropy would be in order.

2) You never quantify how much of the total variation your QTL explains.

3) You never quantify how much of each QTL your identified causal SNPs explain.

4) When dissecting your UBR1 alleles, would be nice to pay homage to this classic:

Laurie-Ahlberg, C. C., and L. F. Stam. 1987. "Use of P-Element-Mediated Transformation to Identify the Molecular Basis of Naturally Occurring Variants Affecting Adh Expression in *Drosophila melanogaster*." Genetics 115 (1): 129-40.

*Reviewer #2 (Recommendations for the authors):*

The only thing I would have appreciated is the validation of some of their findings using orthogonal approaches. There are a lot of readily established biochemical assays they can use to support their claims.

*Reviewer #3 (Recommendations for the authors):*

1. Figure 1 was super helpful in explaining the question and the methods used in this paper.

2. In this sentence, I am not sure that the second portion follows from the first: "These results are consistent with our observation that RM had higher UPS activity for 15 of 20 N-degrons (Figure 1D, Supplementary Table 1), suggesting that the QTLs we have mapped underlie a substantial portion of the heritable UPS activity difference between BY and RM." Maybe you missed the majority of the genetic variation that influences UPS activity, but you got lucky and the portion you detected just happened to be consistent with the overall trend. I think the authors need a different type of analysis to draw a conclusion about whether they sampled "a substantial portion" of heritable differences or whether there is missing heritability here. Can the authors use their data to calculate the narrow sense heritability and then report how much of that heritability is explained by the detected variants?

3. Perhaps reconsider using the term N-degron in the abstract. It's a bit specific. The intro and figures do a great job at defining an N-degron.

4. I did not know that each of the 20 amino acids is dealt with by one of two UPS systems. This was eventually clear in figure 1B where it seems Trp and Met N-degrons are degraded by different systems. Perhaps make this clearer in the text where N-degrons are described and defined.

5. I was mildly confused by the "italics vs plain" statement in figure 3D. Is it intended for only this panel or for the whole figure, or for the whole paper?

6. Some of the focus of the manuscript might be shifted away from general "straw man" questions, like whether this trait is mendelian and controlled by large effect variants (intro lines 55 – 66). The discovery of smaller effect variants is presented as a major finding, but it seems obvious that these exist. Perhaps instead focus on a more quantitative analysis of the power of the current study. How much heritability is explained by previously known genes or variants, and how much additional heritability is explained by the previously unknown genes/variants detected here? Even if a heritability analysis is not feasible, it think a shift in focus from a qualitative statement – we detect small effect variants – to a more quantitative or nuanced statement, would be appropriate.

*Reviewer #4 (Recommendations for the authors):*

I'm interested in the overall pattern that BY seems to have systematically lower UPS activity than RM. BY carries a rare allele at 3 out of the 4 examples in Figure S5, which hints that perhaps there has been an adaptation of BY for lower UPS. It would be interesting to explore this hypothesis further, or at least to discuss it. Has there been an overall adaptation for BY to be less transcriptionally active, coupled with lower degradation rates? Perhaps this might be explored using the previous data sets on mRNA in these lines – presumably, changes in transcription under this model could be either cis or trans. (That analysis is distinct from the analysis reported at L312, which focuses on specific trans-effects of the UPS variants.) And perhaps the authors may have other ideas as well to explore the evolutionary questions.

The paper reports 149 loci, but if I am reading this correctly it looks like this may double-count shared loci. It would be nice to discuss a bit more about likely sharing (ie pleiotropy) of the hits. (It's also a bit hard to see from 2B which of these are likely shared.)

L143: it should be possible to make this statement quantitative.

Para 303, 312: It seems likely that you may be looking for an effect that is small at individual genes but widespread. I would suggest looking more carefully at the overall distribution: eg in the protein data is there an upward shift in the mean? You could also use a more sophisticated method like ashR to study the distribution of changes. For Figure 5 you could also consider adding information about global patterns in means and correlations, which are difficult to read from these plots.

---

## [Author Response]

Reviewer #1 (Recommendations for the authors):I trust more mechanistic reviewers to comment on your experiments. From my point of view, I have only a few suggestions for improvements:1) You argue (in connection with Figure 2)1 that "many QTLs were found only for individual pathways", but this is surely sensitive to significance thresholds. A more sophisticated analysis of pleiotropy would be in order.

The revised manuscript includes an expanded analysis of pleiotropy and how the significance threshold used for QTL detection influences QTL pathway specificity. The high degree of QTL pathway specificity we observed was robust to these additional analyses.

As described in the revised manuscript (pg. 6, para. 3, line 180), we considered a QTL's effect common to multiple N-degrons when QTL peaks for distinct N-degrons were within 100 kb of each other and had the same direction of effect. Using these criteria, the 149 QTLs detected for the 20 N-degrons correspond to 35 distinct QTL regions that each affect between 1 and 11 N-degrons. At the LOD score threshold of 4.5 used in the manuscript, 23 of these 35 QTL regions specifically affected N-degrons from an individual pathway in the N-end Rule, with 11 regions affecting only Ac/N-degrons and 12 regions affecting only Arg/N-degrons. Five of the 23 distinct, pathway-specific QTL regions exclusively affected individual N-degrons. The remaining 12 of the 35 distinct QTL regions have the same direction of effect for at least one reporter each from the Arg/N-end and Ac/N-end pathways. The revised manuscript includes additional text in the Results section (pg. 6, para. 3) discussing pleiotropy among QTLs and a new table (Figure 2—figure supplement 2), which provides detailed information on the 35 distinct QTL regions, including their pathway- and N-degron specificity.

To understand the influence of significance threshold on these results, we analyzed how relaxing the LOD score threshold affected the number of pathway- or N-degron-specific QTL regions. The results of this analysis are summarized in Author response table 1. Supplementary file 2 shows the LOD score and allele frequency difference traces for all 20 N-degrons at the 23 N-end Rule pathway- or N-degron-specific QTL regions at multiple significance thresholds. Author response table 2 and Supplementary file 2 provide detailed information on how the N-end Rule pathway- and N-degron-specificity of each of the 35 distinct QTL regions is influenced by LOD score threshold.

**Author response table 1. sa2table1:** Influence of LOD score significance threshold on the fraction of N-end rule pathway- and N-degron-specific QTLs.

	LOD 4.5	LOD 3.5	LOD 2.5
N-end Rule Pathway-specific QTLs	23 / 35 (66%)	19 / 35 (54%)	16 / 35 (46%)
N-degron-specific QTLs	5 / 35 (14%)	3 / 35 (9%)	3 / 35 (9%)

**Author response table 2. sa2table2:** Shared QTL Regions and N-degrons Affected.

N	QTL	chr	QTL_CI_left	QTL_peak	QTL_CI_right	LOD	RM_AFD	N. N-degrons Affected	N-degrons Affected	Pathway-Specific (LOD 4.5)	Pathway-Specific (LOD 3.5)	Pathway-Specific (LOD 2.5)	N-degron-Specific (LOD 4.5)	N-degron-Specific (LOD 3.5)	N-degron-Specific (LOD 2.5)
1	chr_1_a	1	5021	39400	52122	45	-0.325	7	Ala, Cys, Gly, Pro, Ser, Thr, Val	Ac/N-end	Ac/N-end	Ac/N-end	no	no	no
2	chr_2_a	2	473450	515516	573934	11.1	0.13	3	Arg, Lys, Phe	Arg/N-end	Arg/N-end	no	no	no	no
3	chr_2_b	2	534970	585630	640010	8.9	0.112	5	Ala, Arg, Asp, Lys, Met	no	no	no	no	no	no
4	chr_4_a	4	29500	87662	116525	7.3	-0.108	4	Gln, Glu, Gly, His	no	no	no	no	no	no
5	chr_4_b	4	273175	327300	362600	17.5	0.152	2	Trp, Tyr	Arg/N-end	Arg/N-end	no	no	no	no
6	chr_4_c	4	376817	425550	463567	14.3	0.156	6	Ala, Phe, Pro, Ser, Thr, Trp	no	no	no	no	no	no
7	chr_4_d	4	392275	498200	557000	8.2	-0.107	2	Asn, Asp	Arg/N-end	Arg/N-end	Arg/N-end	no	no	no
8	chr_5_a	5	351400	372136	396250	37.5	0.248	7	Ala, Cys, Gly, Pro, Ser, Thr, Val	Ac/N-end	Ac/N-end	Ac/N-end	no	no	no
13	chr_7_a	7	42075	62775	103225	12.5	-0.155	2	Asn, Thr	no	no	no	no	no	no
9	chr_7_b	7	98178	132779	172279	14.3	-0.156	7	Ala, Cys, Gly, Pro, Ser, Thr, Val	Ac/N-end	no	no	no	no	no
10	chr_7_c	7	389941	418641	451318	18	0.167	11	Ala, Asn, Cys, Gly, Phe, Pro, Ser, Thr, Trp, Tyr, Val	no	no	no	no	no	no
11	chr_7_d	7	841300	869700	899934	15.2	-0.157	3	Arg, Asn, Asp	Arg/N-end	Arg/N-end	Arg/N-end	no	no	no
12	chr_7_e	7	856120	870980	883830	107.5	0.409	5	His, Leu, Phe, Trp, Tyr	Arg/N-end	Arg/N-end	Arg/N-end	no	no	no
15	chr_8_a	8	50150	98050	127300	6	-0.101	1	Tyr	Arg/N-end	Arg/N-end	Arg/N-end	Tyr	Tyr	Tyr
14	chr_8_b	8	118875	148400	193675	8.4	0.104	2	Asp, Lys	Arg/N-end	Arg/N-end	Arg/N-end	no	no	no
17	chr_9_a	9	94550	130775	168000	5.1	0.099	2	His, Lys	Arg/N-end	Arg/N-end	Arg/N-end	no	no	no
16	chr_9_b	9	267641	291492	315417	23.5	0.195	6	Ala, Gly, Pro, Ser, Thr, Val	Ac/N-end	Ac/N-end	Ac/N-end	no	no	no
18	chr_10_a	10	323900	345482	371186	21.9	0.182	11	Ala, Arg, Cys, Gln, Gly, Lys, Phe, Pro, Ser, Trp, Tyr	no	no	no	no	no	no
19	chr_10_b	10	589770	615660	641600	25.1	0.192	5	Ala, Arg, Asn, Cys, Gln	no	no	no	no	no	no
21	chr_11_a	11	118750	156450	173600	16	0.152	1	His	Arg/N-end	Arg/N-end	Arg/N-end	His	no	no
20	chr_11_b	11	291030	339790	388750	8.4	-0.106	5	Ala, Arg, Cys, Phe, Thr	no	no	no	no	no	no
22	chr_12_a	12	157200	197050	226850	8.2	0.126	2	Ala, Pro	Ac/N-end	no	no	no	no	no
23	chr_12_b	12	644414	657241	674723	42.1	-0.25	11	Ala, Cys, Gly, Met, Phe, Pro, Ser, Thr, Trp, Tyr, Val	no	no	no	no	no	no
24	chr_12_c	12	639637	674125	702650	9.7	0.108	4	Asp, Gly, His, Ile	Arg/N-end	Arg/N-end	Arg/N-end	no	no	no
25	chr_13_a	13	0	24650	58950	9.6	-0.162	1	Ala	Ac/N-end	Ac/N-end	no	Ala	no	no
27	chr_13_b	13	31800	56525	77675	14.9	0.145	2	His, Phe	Arg/N-end	Arg/N-end	Arg/N-end	no	no	no
26	chr_13_c	13	265350	296167	331584	16.5	0.17	6	Arg, Asn, Asp, Glu, His, Lys	Arg/N-end	no	no	no	no	no
29	chr_14_a	14	450731	465394	482912	89.3	-0.342	8	His, Leu, Lys, Met, Phe, Pro, Trp, Tyr	no	no	no	no	no	no
28	chr_14_b	14	449725	470850	504025	86.7	0.336	2	Asp, Thr	no	no	no	no	no	no
31	chr_15_a	15	133625	163862	184831	20.2	-0.183	8	Arg, Asn, Asp, Glu, His, Lys, Met, Phe	no	no	no	no	no	no
30	chr_15_b	15	342850	388050	431300	10.1	0.116	3	Ala, Pro, Thr	Ac/N-end	no	no	no	no	no
32	chr_15_c	15	525200	561525	591775	10.6	0.128	2	Ser, Thr	Ac/N-end	Ac/N-end	Ac/N-end	no	no	no
33	chr_15_d	15	518550	569750	594650	8.8	-0.096	1	Cys	Ac/N-end	Ac/N-end	Ac/N-end	Cys	Cys	Cys
34	chr_16_a	16	166030	194830	222070	11.6	0.121	5	Ala, Gly, Pro, Thr, Val	Ac/N-end	Ac/N-end	Ac/N-end	no	no	no
35	chr_16_b	16	375050	403350	428950	24.5	-0.22	1	Pro	Ac/N-end	Ac/N-end	Ac/N-end	Pro	Pro	Pro

Abbreviations: "chr": chromosome, "CI": confidence interval, "RM_AFD": RM allele frequency difference (high – low UPS activity pools).

Generally, altering the significance threshold does not affect the conclusions that at least half of the detected QTLs are specific to an individual N-end Rule pathway and that approximately 10% of QTLs are specific to individual N-degrons. In the revised Results section (pg. 6, para. 3, line 18), the qualitative statement mentioned by the reviewer is replaced with the quantitative information in Author response table 1 for the 4.5 LOD threshold column.

We note that N-end Rule pathway-specificity for 3 QTL regions (IVb, IXa, and XVc) is subject to the following considerations that may not be immediately obvious from reviewing Supplementary file 2.

The Arg/N-end-specific chromosome IVb and IXa QTL regions overlap the leftmost shoulders of QTLs detected with Ac/N-degrons, creating the appearance that these QTLs are not Arg/N-degron-specific when plotted as in Supplementary file 2. However, the peaks of these Arg/N-degron and Ac/N-degron QTLs are not within 100 kb and their confidence intervals do not overlap. We therefore interpret their effects as pathway-specific.

The Ac/N-degron-specific chromosome XVc QTL contains peaks from both glutamate Arg/N-degron replicates. However, the direction of effect of these two peaks is not consistent between replicates (the only such instance in our dataset). Because we could not unambiguously assign a direction of effect for the glutamate N-degron at this region, we do not include it in our set of QTLs and the chromosome XVc region is deemed to be Ac/N-degron-specific. In the revised manuscript, we report the chromosome IVb, IXa, and XVc QTL regions as pathway-specific.

We also note that the pathway-specific QTL regions displayed on pages 7 and 8 (QTL regions containing UBR1) as well as 20 and 21 of Supplementary file 2 are highly overlapping. However, these overlapping regions have opposing directions of effect on the sets of reporters they affect. We, therefore, continue to report these QTL regions as distinct in the revised manuscript, an interpretation supported by our fine-mapping data (see e.g., Figure 3C).

2) You never quantify how much of the total variation your QTL explains.

Our bulk segregant analysis QTL mapping method is based on comparing allele frequencies obtained from whole-genome sequencing pools of cells with extreme phenotypes. The genotypes of individual segregants, which would be needed for calculating explained variance, are not ascertained using this approach. Thus, it is not readily possible to calculate the proportion of variance explained by our QTLs.

3) You never quantify how much of each QTL your identified causal SNPs explain.

Author response table 3 displays the variance in ubiquitin-proteasome system (UPS) activity explained by each tested causal gene allele and variant.

**Author response table 3. sa2table3:** Variance Explained by Causal Alleles and Variants.

Gene	Allele	N-degron	Variance Explained
*DOA10*	K1012N	Thr	0.04
*DOA10*	Q410E	Thr	0.075
*DOA10*	RM_full	Thr	0.911
*DOA10*	Y1186F	Thr	0.597
*DOA10*	K1012N	Gly	0.608
*DOA10*	Q410E	Gly	0.74
*DOA10*	RM_full	Gly	0.879
*DOA10*	Y1186F	Gly	0.77
*NTA1*	D111E	Asn	0.204
*NTA1*	E129G	Asn	0.764
*NTA1*	RM_full	Asn	0.789
*NTA1*	RM_pr	Asn	-0.014
*UBC6*	D229G	Ala	0.717
*UBC6*	RM_full	Ala	0.666
*UBC6*	RM_pr	Ala	-0.027
*UBC6*	RM_term	Ala	0.183
*UBC6*	D229G	Thr	0.762
*UBC6*	RM_full	Thr	0.928
*UBC6*	RM_pr	Thr	0.008
*UBC6*	RM_term	Thr	0.006
*UBR1*	RM_causal	Phe	0.9
*UBR1*	RM_full	Phe	0.917
*UBR1*	RM_non	Phe	-0.019
*UBR1*	RM_pr	Phe	0.818
*UBR1*	RM_causal	Trp	0.937
*UBR1*	RM_full	Trp	0.975
*UBR1*	RM_non	Trp	0.059
*UBR1*	RM_pr	Trp	0.934
*UBR1*	RM_full	Asn	0.438
*UBR1*	RM_ORF	Asn	0.134
*UBR1*	RM_pr	Asn	-0.012
*UBR1*	RM_term	Asn	0.064
*UBR1*	RM_full	Asp	0.403
*UBR1*	RM_ORF	Asp	0.101
*UBR1*	RM_pr	Asp	0.097
*UBR1*	RM_term	Asp	0.048
*UBR1*	RM_full	Phe	0.866
*UBR1*	RM_ORF	Phe	0.343
*UBR1*	RM_pr	Phe	0.316
*UBR1*	RM_term	Phe	-0.031
*UBR1*	RM_full	Trp	0.964
*UBR1*	RM_ORF	Trp	0.813
*UBR1*	RM_pr	Trp	0.928
*UBR1*	RM_term	Trp	0.028

We note that these estimates are obtained from experiments in near-isogenic strains that differ only at the tested causal gene allele or variant. The fraction of variance explained is thus inflated relative to what would be observed in the segregant populations used for QTL mapping and should not be used to estimate the variance explained by a QTL region. Therefore, we do not include these estimates in the revised manuscript.

4) When dissecting your UBR1 alleles, would be nice to pay homage to this classic: Laurie-Ahlberg, C. C., and L. F. Stam. 1987. "Use of P-Element-Mediated Transformation to Identify the Molecular Basis of Naturally Occurring Variants Affecting Adh Expression in *Drosophila melanogaster*." Genetics 115 (1): 129-40.

We thank the reviewer for the suggestion to include this important work on identifying causal variants for enzyme activity in highly polymorphic genomic regions. The work appears as reference 62 (pg. 8, para 3, line 218) in the revised manuscript.

Reviewer #2 (Recommendations for the authors): The only thing I would have appreciated is the validation of some of their findings using orthogonal approaches. There are a lot of readily established biochemical assays they can use to support their claims.

Previously published theoretical and empirical observations have demonstrated that tandem fluorescent timers (TFTs) provide precise and sensitive measures of protein degradation kinetics. In particular, the TFT system has been extensively used to measure differences in the degradation rate of UPS N-end Rule substrates (Khmelinskii et al., 2012, Khmelinskii and Knop, 2014, Kats et al., 2018). Given the well-established validity of the TFT system for measuring N-end Rule activity and the comparatively lower precision, sensitivity, and throughput of conventional biochemical measurements of protein degradation (in particular, pulse-chase Western blotting and cycloheximide chase analysis [Kong et al., 2021]), we argue that further experimentation is not needed to establish the claims made in our work.

Reviewer #3 (Recommendations for the authors):1. Figure 1 was super helpful in explaining the question and the methods used in this paper.

Thank you!

2. In this sentence, I am not sure that the second portion follows from the first: "These results are consistent with our observation that RM had higher UPS activity for 15 of 20 N-degrons (Figure 1D, Supplementary Table 1), suggesting that the QTLs we have mapped underlie a substantial portion of the heritable UPS activity difference between BY and RM." Maybe you missed the majority of the genetic variation that influences UPS activity, but you got lucky and the portion you detected just happened to be consistent with the overall trend. I think the authors need a different type of analysis to draw a conclusion about whether they sampled "a substantial portion" of heritable differences or whether there is missing heritability here. Can the authors use their data to calculate the narrow sense heritability and then report how much of that heritability is explained by the detected variants?

As noted in our response to Reviewer 1, because of the pooled nature of our genetic mapping method, it is not readily possible to calculate heritability from our QTL mapping data. Author response table 3 presents the variance explained by the tested causal alleles and variants. However, as noted in our response to Reviewer 1, these estimates are inflated because they are derived from near-isogenic cell populations that differ only at the tested alleles / variants. Given these considerations, the second clause in the sentence mentioned by the reviewer has been removed from the revised manuscript.

3. Perhaps reconsider using the term N-degron in the abstract. It's a bit specific. The intro and figures do a great job at defining an N-degron.

The term “N-degron” has been removed from the abstract and replaced with more general terms.

4. I did not know that each of the 20 amino acids is dealt with by one of two UPS systems. This was eventually clear in figure 1B where it seems Trp and Met N-degrons are degraded by different systems. Perhaps make this clearer in the text where N-degrons are described and defined.

We thank the reviewer for this suggestion to improve the manuscript’s clarity. The revised introduction indicates that the N-end Rule contains two distinct targeting complexes when introducing the N-end Rule (pg. 3, para. 2, line 105) and references Figure 1, which illustrates the two pathways of the N-end rule.

5. I was mildly confused by the "italics vs plain" statement in figure 3D. Is it intended for only this panel or for the whole figure, or for the whole paper?

The “italics vs. plain” statement is intended only for panels 3D and 4B/F/J. To improve clarity, we have added additional annotation to these panels.

6. Some of the focus of the manuscript might be shifted away from general "straw man" questions, like whether this trait is mendelian and controlled by large effect variants (intro lines 55 – 66). The discovery of smaller effect variants is presented as a major finding, but it seems obvious that these exist. Perhaps instead focus on a more quantitative analysis of the power of the current study. How much heritability is explained by previously known genes or variants, and how much additional heritability is explained by the previously unknown genes/variants detected here? Even if a heritability analysis is not feasible, it think a shift in focus from a qualitative statement – we detect small effect variants – to a more quantitative or nuanced statement, would be appropriate.

The revised manuscript provides a more nuanced introduction to the genetics of UPS activity, in particular, emphasizing the expectation that UPS activity, like most traits, is genetically complex. We agree with the reviewer that the relative contributions of individual QTLs to the heritability of UPS activity would be an interesting and informative analysis. However, as noted in our response to Reviewer 1, it is not readily feasible to perform such an analysis. Instead, as suggested by the reviewer, several qualitative statements have been replaced by quantitative descriptions. In particular, the qualitative statement mentioned by the reviewer is replaced with a quantitative statement regarding QTL effect sizes (pg. 6, para. 1, line 164).

Reviewer #4 (Recommendations for the authors):Specific comments:I'm interested in the overall pattern that BY seems to have systematically lower UPS activity than RM. BY carries a rare allele at 3 out of the 4 examples in Figure S5, which hints that perhaps there has been an adaptation of BY for lower UPS. It would be interesting to explore this hypothesis further, or at least to discuss it. Has there been an overall adaptation for BY to be less transcriptionally active, coupled with lower degradation rates? Perhaps this might be explored using the previous data sets on mRNA in these lines – presumably, changes in transcription under this model could be either cis or trans. (That analysis is distinct from the analysis reported at L312, which focuses on specific trans-effects of the UPS variants.) And perhaps the authors may have other ideas as well to explore the evolutionary questions.

We thank the reviewer for these interesting suggestions, which we have addressed with a series of new analyses. In brief, we did not detect evidence for lineage-specific selection on UPS gene expression using eQTL data or on individual N-end reporters.

To explore whether the consistent differences in UPS activity between the BY and RM strains might reflect adaptive changes in these lineages, we performed several additional analyses. We first applied the sign test (Fraser et al., 2010; https://doi.org/10.1073/pnas.0912245107) to a recent comprehensive BY / RM eQTL mapping dataset, which became available after the original sign test publication. This newer eQTL dataset comprises 36,498 eQTLs mapped for 5,643 genes in a panel of 1,000 recombinant offspring from the BY / RM cross (Albert et al., 2018; https://doi.org/10.7554/eLife.35471). The results of this analysis are presented in Author response table 4. We performed the analysis at 11 different LOD thresholds (ranging from 2.5 to 50) to examine the influence of QTL effect size on the results. Across all genes, we do not find evidence for lineage-specific selection except at LOD thresholds of 40 and 45. Given the large fraction of eQTLs that are excluded at these high thresholds (94.6% and 95.3%), the large fraction of genes excluded (70.5% and 73.8%), and especially the marginally significant p-values (0.038 and 0.026) obtained at these two thresholds, we conclude that there is, at best, limited evidence from the sign test for lineage-specific selection on overall mRNA transcript abundance in the BY / RM cross. Future work is needed to reconcile discrepancies in the results of the sign test as obtained in different eQTL datasets from the same cross.

**Author response table 4. sa2table4:** Results of the Sign Test for Lineage-Specific Selection Applied to All BY / RM eQTLs.

LOD	n_eQTLs	n_genes	n_pairs	reinf_BY_up	reinf_RM_up	oppos_BY_up	oppos_RM_up	excess_reinforcing_pairs	chi_sq_p
2.4	36498	5643	2845	627	757	952	509	–14	0.818
5	19694	5372	2081	449	577	698	357	19	0.701
10	9379	4514	1124	225	320	404	175	4.63	0.934
15	6125	3686	745	154	202	279	110	2.24	0.992
20	4452	3076	480	108	128	179	65	18.2	0.431
25	3458	2609	336	73	95	127	41	20.6	0.276
30	2788	2224	230	54	65	85	26	22.6	0.145
35	2310	1912	178	44	48	68	18	20	0.144
40	1963	1664	140	35	41	53	11	24.3	0.0375
45	1706	1474	110	28	33	42	7	22.9	0.0261
50	1475	1299	85	21	27	31	6	17.9	0.0569

Abbreviations: “LOD”: LOD score threshold for calling cis / trans eQTL pairs, "n_eQTLs": number of eQTLs, "n_genes": number of genes with an eQTL, "n_pairs": number of cis / trans eQTL pairs, "reinf_BY_up": number of cis / trans eQTLs where the BY allele of the cis and trans eQTLs increases expression (reinforcing pairs), "reinf_RM_up": number of cis / trans eQTLs where the RM allele of the cis and trans eQTLs increases expression (reinforcing pairs), "oppos_BY_up": number of cis / trans eQTLs where the BY allele of the cis eQTL increases expression and the RM allele of the trans eQTL increases expression (opposing pairs), "oppos_RM_up": number of cis / trans eQTLs where the RM allele of the cis eQTL increases expression and the BY allele of the trans eQTL increases expression (opposing pairs), "excess_reinforcing_pairs", the number of excess reinforcing eQTL pairs calculated as in Fraser et al., 2010, "chi_sq_p": p-value of the chi-square test for enrichment of reinforcing pairs.

As in the original manuscript sign test manuscript, we performed a gene ontology (GO) enrichment analysis on the sets of reinforcing cis / trans eQTL pairs from the set of eQTLs to determine whether such pairs are enriched for UPS genes. We were able to replicate the previously-described enrichment for genes of the ergosterol biosynthesis pathway (Fraser et al., 2010, Author response table 5). However, there was no enrichment for ubiquitin system or proteasome GO terms in the sets of reinforcing eQTL pairs at any of the tested LOD score thresholds (Author response table 5).

**Author response table 5. sa2table5:** Results of Gene Ontology Enrichment of All cis / trans eQTL pairs.

GOBPID	p_value	OddsRatio	ExpCount	Count	Term	category	LOD
GO:0032197	2.8E-05	3	15.06	30	transposition; RNA-mediated	all_reinforcing	2.426606803
GO:0015074	0.00046	3.1	9.87	20	DNA integration	all_reinforcing	2.426606803
GO:0006487	0.00077	2.7	11.6	22	protein N-linked glycosylation	all_reinforcing	2.426606803
GO:0046513	0.0012	10.7	2.22	7	ceramide biosynthetic process	all_reinforcing	2.426606803
GO:0006458	0.0029	3.3	6.17	13	'de novo' protein folding	all_reinforcing	2.426606803
GO:0018904	0.0037	Inf	0.99	4	ether metabolic process	all_reinforcing	2.426606803
GO:0051131	0.0039	9.2	1.97	6	chaperone-mediated protein complex assembly	all_reinforcing	2.426606803
GO:1901135	0.0045	1.4	92.08	114	carbohydrate derivative metabolic process	all_reinforcing	2.426606803
GO:0070085	0.0063	1.8	21.23	32	glycosylation	all_reinforcing	2.426606803
GO:0061077	0.0068	3.9	3.95	9	chaperone-mediated protein folding	all_reinforcing	2.426606803
GO:0006672	0.007	5.4	2.72	7	ceramide metabolic process	all_reinforcing	2.426606803
GO:0009101	0.007	1.8	19.75	30	glycoprotein biosynthetic process	all_reinforcing	2.426606803
GO:0016226	0.0089	3.1	5.43	11	iron-sulfur cluster assembly	all_reinforcing	2.426606803
GO:0031070	0.0093	6.1	2.22	6	intronic snoRNA processing	all_reinforcing	2.426606803
GO:0034965	0.0093	6.1	2.22	6	intronic box C/D snoRNA processing	all_reinforcing	2.426606803
GO:0032197	2.8E-05	3	15.06	30	transposition; RNA-mediated	all_reinforcing	2.426606803
GO:0015074	0.00046	3.1	9.87	20	DNA integration	all_reinforcing	2.426606803
GO:0006487	0.00077	2.7	11.6	22	protein N-linked glycosylation	all_reinforcing	2.426606803
GO:0046513	0.0012	10.7	2.22	7	ceramide biosynthetic process	all_reinforcing	2.426606803
GO:0006458	0.0029	3.3	6.17	13	'de novo' protein folding	all_reinforcing	2.426606803
GO:0018904	0.0037	Inf	0.99	4	ether metabolic process	all_reinforcing	2.426606803
GO:0051131	0.0039	9.2	1.97	6	chaperone-mediated protein complex assembly	all_reinforcing	2.426606803
GO:1901135	0.0045	1.4	92.08	114	carbohydrate derivative metabolic process	all_reinforcing	2.426606803
GO:0070085	0.0063	1.8	21.23	32	glycosylation	all_reinforcing	2.426606803
GO:0061077	0.0068	3.9	3.95	9	chaperone-mediated protein folding	all_reinforcing	2.426606803
GO:0006672	0.007	5.4	2.72	7	ceramide metabolic process	all_reinforcing	2.426606803
GO:0009101	0.007	1.8	19.75	30	glycoprotein biosynthetic process	all_reinforcing	2.426606803
GO:0016226	0.0089	3.1	5.43	11	iron-sulfur cluster assembly	all_reinforcing	2.426606803
GO:0031070	0.0093	6.1	2.22	6	intronic snoRNA processing	all_reinforcing	2.426606803
GO:0034965	0.0093	6.1	2.22	6	intronic box C/D snoRNA processing	all_reinforcing	2.426606803
GO:0032197	4E-06	3.5	11.51	27	transposition; RNA-mediated	all_reinforcing	5
GO:0015074	0.00011	3.7	7.48	18	DNA integration	all_reinforcing	5
GO:0006458	0.001	3.9	4.79	12	'de novo' protein folding	all_reinforcing	5
GO:0061077	0.0011	5.5	3.07	9	chaperone-mediated protein folding	all_reinforcing	5
GO:0006487	0.0013	2.7	8.82	18	protein N-linked glycosylation	all_reinforcing	5
GO:0018904	0.0013	Inf	0.77	4	ether metabolic process	all_reinforcing	5
GO:0046513	0.0024	8.5	1.73	6	ceramide biosynthetic process	all_reinforcing	5
GO:0070525	0.0024	8.5	1.73	6	tRNA threonylcarbamoyladenosine metabolic process	all_reinforcing	5
GO:0051131	0.0038	10.6	1.34	5	chaperone-mediated protein complex assembly	all_reinforcing	5
GO:0032259	0.0045	1.8	21.29	33	methylation	all_reinforcing	5
GO:0046165	0.0052	2.2	11.32	20	alcohol biosynthetic process	all_reinforcing	5
GO:0044281	0.0063	1.3	150.75	177	small molecule metabolic process	all_reinforcing	5
GO:0006662	0.007	Inf	0.58	3	glycerol ether metabolic process	all_reinforcing	5
GO:0002949	0.007	Inf	0.58	3	tRNA threonylcarbamoyladenosine modification	all_reinforcing	5
GO:0033215	0.007	Inf	0.58	3	iron assimilation by reduction and transport	all_reinforcing	5
GO:0046131	0.0083	3.8	3.26	8	pyrimidine ribonucleoside metabolic process	all_reinforcing	5
GO:0019509	0.0086	7.1	1.53	5	L-methionine salvage from methylthioadenosine	all_reinforcing	5
GO:0006694	0.0086	2.4	8.06	15	steroid biosynthetic process	all_reinforcing	5
GO:0006696	0.0089	2.7	5.95	12	ergosterol biosynthetic process	all_reinforcing	5
GO:0006672	0.0094	5.1	2.11	6	ceramide metabolic process	all_reinforcing	5
GO:0019856	0.0094	5.1	2.11	6	pyrimidine nucleobase biosynthetic process	all_reinforcing	5
GO:1902652	0.0098	2.5	6.71	13	secondary alcohol metabolic process	all_reinforcing	5
GO:0032197	1.5E-06	4.5	6.44	20	transposition; RNA-mediated	all_reinforcing	10
GO:0015074	7.7E-05	4.3	4.62	14	DNA integration	all_reinforcing	10
GO:0018904	0.00022	Inf	0.49	4	ether metabolic process	all_reinforcing	10
GO:0006662	0.0018	Inf	0.36	3	glycerol ether metabolic process	all_reinforcing	10
GO:0006278	0.0021	2.7	6.8	15	RNA-dependent DNA biosynthetic process	all_reinforcing	10
GO:1902047	0.0022	9.1	1.09	5	polyamine transmembrane transport	all_reinforcing	10
GO:0046165	0.0041	2.6	6.56	14	alcohol biosynthetic process	all_reinforcing	10
GO:0043605	0.0056	9.7	0.85	4	cellular amide catabolic process	all_reinforcing	10
GO:0019509	0.0056	9.7	0.85	4	L-methionine salvage from methylthioadenosine	all_reinforcing	10
GO:0072488	0.006	4.9	1.82	6	ammonium transmembrane transport	all_reinforcing	10
GO:0015846	0.0064	6.1	1.34	5	polyamine transport	all_reinforcing	10
GO:0006833	0.0065	21.8	0.49	3	water transport	all_reinforcing	10
GO:0042044	0.0065	21.8	0.49	3	fluid transport	all_reinforcing	10
GO:1902652	0.007	2.9	4.25	10	secondary alcohol metabolic process	all_reinforcing	10
GO:0002098	0.007	3.9	2.43	7	tRNA wobble uridine modification	all_reinforcing	10
GO:0006400	0.008	2.3	7.05	14	tRNA modification	all_reinforcing	10
GO:0009067	0.0086	2.5	5.71	12	aspartate family amino acid biosynthetic process	all_reinforcing	10
GO:0006696	0.0092	3	3.77	9	ergosterol biosynthetic process	all_reinforcing	10
GO:0006694	0.0096	2.6	5.1	11	steroid biosynthetic process	all_reinforcing	10
GO:0034220	0.0096	1.6	25.27	37	ion transmembrane transport	all_reinforcing	10
GO:0032259	0.01	1.9	11.66	20	methylation	all_reinforcing	10
GO:0015074	3.3E-05	5	3.71	13	DNA integration	all_reinforcing	15
GO:0032197	6.2E-05	4	4.98	15	transposition; RNA-mediated	all_reinforcing	15
GO:0046165	9.3E-05	4	4.59	14	alcohol biosynthetic process	all_reinforcing	15
GO:0006278	0.00031	3.5	5.07	14	RNA-dependent DNA biosynthetic process	all_reinforcing	15
GO:1902652	0.0011	3.9	3.32	10	secondary alcohol metabolic process	all_reinforcing	15
GO:0019509	0.0011	18.7	0.59	4	L-methionine salvage from methylthioadenosine	all_reinforcing	15
GO:0070525	0.0011	18.7	0.59	4	tRNA threonylcarbamoyladenosine metabolic process	all_reinforcing	15
GO:0006696	0.0016	4	2.93	9	ergosterol biosynthetic process	all_reinforcing	15
GO:0009086	0.0016	4	2.93	9	methionine biosynthetic process	all_reinforcing	15
GO:0071267	0.0025	12.5	0.68	4	L-methionine salvage	all_reinforcing	15
GO:0043102	0.0025	12.5	0.68	4	amino acid salvage	all_reinforcing	15
GO:0043605	0.0025	12.5	0.68	4	cellular amide catabolic process	all_reinforcing	15
GO:0016128	0.0033	3.5	3.22	9	phytosteroid metabolic process	all_reinforcing	15
GO:0006811	0.0034	1.7	27.81	42	ion transport	all_reinforcing	15
GO:0070900	0.0034	28	0.39	3	mitochondrial tRNA modification	all_reinforcing	15
GO:1900864	0.0034	28	0.39	3	mitochondrial RNA modification	all_reinforcing	15
GO:0006694	0.0049	3	4	10	steroid biosynthetic process	all_reinforcing	15
GO:0009066	0.005	2.6	5.95	13	aspartate family amino acid metabolic process	all_reinforcing	15
GO:0044107	0.0051	3.3	3.42	9	cellular alcohol metabolic process	all_reinforcing	15
GO:0016125	0.0063	2.7	4.78	11	sterol metabolic process	all_reinforcing	15
GO:1902047	0.0075	7.5	0.88	4	polyamine transmembrane transport	all_reinforcing	15
GO:0006531	0.0079	14	0.49	3	aspartate metabolic process	all_reinforcing	15
GO:0090646	0.0079	14	0.49	3	mitochondrial tRNA processing	all_reinforcing	15
GO:0015804	0.0082	5.2	1.37	5	neutral amino acid transport	all_reinforcing	15
GO:0016226	0.0082	5.2	1.37	5	iron-sulfur cluster assembly	all_reinforcing	15
GO:0098655	0.0093	1.9	12.29	21	cation transmembrane transport	all_reinforcing	15
GO:0015840	0.0095	Inf	0.2	2	urea transport	all_reinforcing	15
GO:0034311	0.0095	Inf	0.2	2	diol metabolic process	all_reinforcing	15
GO:0034312	0.0095	Inf	0.2	2	diol biosynthetic process	all_reinforcing	15
GO:0019755	0.0095	Inf	0.2	2	one-carbon compound transport	all_reinforcing	15
GO:0042883	0.0095	Inf	0.2	2	cysteine transport	all_reinforcing	15
GO:0090502	0.0096	2.2	7.12	14	RNA phosphodiester bond hydrolysis; endonucleolytic	all_reinforcing	15
GO:0015074	1.8E-06	6.8	2.87	13	DNA integration	all_reinforcing	20
GO:0032197	2.8E-06	5.4	3.88	15	transposition; RNA-mediated	all_reinforcing	20
GO:0006278	4.4E-05	4.6	3.73	13	RNA-dependent DNA biosynthetic process	all_reinforcing	20
GO:0070525	0.00047	24.2	0.47	4	tRNA threonylcarbamoyladenosine metabolic process	all_reinforcing	20
GO:0090502	0.0006	3.3	4.73	13	RNA phosphodiester bond hydrolysis; endonucleolytic	all_reinforcing	20
GO:0034654	0.0017	1.6	41.44	59	nucleobase-containing compound biosynthetic process	all_reinforcing	20
GO:0006551	0.0017	36.1	0.31	3	leucine metabolic process	all_reinforcing	20
GO:0070900	0.0017	36.1	0.31	3	mitochondrial tRNA modification	all_reinforcing	20
GO:1900864	0.0017	36.1	0.31	3	mitochondrial RNA modification	all_reinforcing	20
GO:0015804	0.0021	7.6	1.01	5	neutral amino acid transport	all_reinforcing	20
GO:0016226	0.0021	7.6	1.01	5	iron-sulfur cluster assembly	all_reinforcing	20
GO:0001302	0.0029	3.9	2.56	8	replicative cell aging	all_reinforcing	20
GO:0044249	0.0039	1.5	91.27	111	cellular biosynthetic process	all_reinforcing	20
GO:1901576	0.004	1.5	92.28	112	organic substance biosynthetic process	all_reinforcing	20
GO:0090646	0.0041	18	0.39	3	mitochondrial tRNA processing	all_reinforcing	20
GO:0009083	0.0041	18	0.39	3	branched-chain amino acid catabolic process	all_reinforcing	20
GO:0006310	0.005	2.2	8.69	17	DNA recombination	all_reinforcing	20
GO:0015840	0.006	Inf	0.16	2	urea transport	all_reinforcing	20
GO:0019755	0.006	Inf	0.16	2	one-carbon compound transport	all_reinforcing	20
GO:0042883	0.006	Inf	0.16	2	cysteine transport	all_reinforcing	20
GO:0043605	0.0077	12	0.47	3	cellular amide catabolic process	all_reinforcing	20
GO:0019509	0.0077	12	0.47	3	L-methionine salvage from methylthioadenosine	all_reinforcing	20
GO:0070880	0.0077	12	0.47	3	fungal-type cell wall beta-glucan biosynthetic process	all_reinforcing	20
GO:0070879	0.0077	12	0.47	3	fungal-type cell wall beta-glucan metabolic process	all_reinforcing	20
GO:0044283	0.0098	1.7	21.19	32	small molecule biosynthetic process	all_reinforcing	20
GO:0015074	1.7E-06	7.4	2.4	12	DNA integration	all_reinforcing	25
GO:0006278	1.7E-05	5.6	2.92	12	RNA-dependent DNA biosynthetic process	all_reinforcing	25
GO:0032197	2.7E-05	5.2	3.05	12	transposition; RNA-mediated	all_reinforcing	25
GO:0090502	9.7E-05	4.5	3.44	12	RNA phosphodiester bond hydrolysis; endonucleolytic	all_reinforcing	25
GO:0034654	0.0014	1.8	29.54	45	nucleobase-containing compound biosynthetic process	all_reinforcing	25
GO:0001302	0.002	4.7	1.88	7	replicative cell aging	all_reinforcing	25
GO:0044249	0.003	1.6	64.61	82	cellular biosynthetic process	all_reinforcing	25
GO:0015804	0.0039	8.4	0.71	4	neutral amino acid transport	all_reinforcing	25
GO:0006310	0.004	2.5	6.43	14	DNA recombination	all_reinforcing	25
GO:1901576	0.004	1.6	65.19	82	organic substance biosynthetic process	all_reinforcing	25
GO:0015840	0.0042	Inf	0.13	2	urea transport	all_reinforcing	25
GO:0019755	0.0042	Inf	0.13	2	one-carbon compound transport	all_reinforcing	25
GO:0070525	0.0046	14.6	0.39	3	tRNA threonylcarbamoyladenosine metabolic process	all_reinforcing	25
GO:0007568	0.0063	3.3	2.86	8	aging	all_reinforcing	25
GO:0090305	0.0087	2.3	6.3	13	nucleic acid phosphodiester bond hydrolysis	all_reinforcing	25
GO:0006696	0.0093	3.9	1.88	6	ergosterol biosynthetic process	all_reinforcing	25
GO:0015074	2E-07	9.3	1.99	12	DNA integration	all_reinforcing	30
GO:0006278	1.6E-06	7.2	2.37	12	RNA-dependent DNA biosynthetic process	all_reinforcing	30
GO:0032197	3.5E-06	6.6	2.53	12	transposition; RNA-mediated	all_reinforcing	30
GO:0090502	4.5E-06	6.4	2.58	12	RNA phosphodiester bond hydrolysis; endonucleolytic	all_reinforcing	30
GO:0006310	0.00028	3.4	4.95	14	DNA recombination	all_reinforcing	30
GO:0090305	0.0019	3	4.68	12	nucleic acid phosphodiester bond hydrolysis	all_reinforcing	30
GO:1902047	0.0045	13.5	0.38	3	polyamine transmembrane transport	all_reinforcing	30
GO:0044249	0.0046	1.7	45	59	cellular biosynthetic process	all_reinforcing	30
GO:0034654	0.0047	1.8	20.59	32	nucleobase-containing compound biosynthetic process	all_reinforcing	30
GO:1901576	0.0059	1.6	45.43	59	organic substance biosynthetic process	all_reinforcing	30
GO:0015846	0.007	10.8	0.43	3	polyamine transport	all_reinforcing	30
GO:1903008	0.0092	4.6	1.34	5	organelle disassembly	all_reinforcing	30
GO:0015074	3.7E-08	11.2	1.74	12	DNA integration	all_reinforcing	35
GO:0032197	4.5E-07	8.4	2.12	12	transposition; RNA-mediated	all_reinforcing	35
GO:0006278	4.5E-07	8.4	2.12	12	RNA-dependent DNA biosynthetic process	all_reinforcing	35
GO:0090502	5.9E-07	8.1	2.17	12	RNA phosphodiester bond hydrolysis; endonucleolytic	all_reinforcing	35
GO:0006310	0.00012	4	4.05	13	DNA recombination	all_reinforcing	35
GO:0090305	0.00021	4	3.71	12	nucleic acid phosphodiester bond hydrolysis	all_reinforcing	35
GO:0044249	0.00062	2.1	34.67	50	cellular biosynthetic process	all_reinforcing	35
GO:0034654	0.00077	2.3	15.62	28	nucleobase-containing compound biosynthetic process	all_reinforcing	35
GO:1901576	0.0008	2	35.01	50	organic substance biosynthetic process	all_reinforcing	35
GO:1902047	0.0033	15.3	0.34	3	polyamine transmembrane transport	all_reinforcing	35
GO:0015846	0.0051	12.2	0.39	3	polyamine transport	all_reinforcing	35
GO:0043457	0.0067	40.3	0.14	2	regulation of cellular respiration	all_reinforcing	35
GO:0006696	0.0095	4.5	1.35	5	ergosterol biosynthetic process	all_reinforcing	35
GO:0015074	1.8E-08	12.2	1.64	12	DNA integration	all_reinforcing	40
GO:0090502	1.7E-07	9.4	1.96	12	RNA phosphodiester bond hydrolysis; endonucleolytic	all_reinforcing	40
GO:0032197	2.2E-07	9.1	2.01	12	transposition; RNA-mediated	all_reinforcing	40
GO:0006278	2.2E-07	9.1	2.01	12	RNA-dependent DNA biosynthetic process	all_reinforcing	40
GO:0006310	3.1E-05	4.8	3.61	13	DNA recombination	all_reinforcing	40
GO:0090305	6.7E-05	4.7	3.33	12	nucleic acid phosphodiester bond hydrolysis	all_reinforcing	40
GO:0015846	0.0017	21.7	0.27	3	polyamine transport	all_reinforcing	40
GO:1902047	0.0017	21.7	0.27	3	polyamine transmembrane transport	all_reinforcing	40
GO:0044249	0.0027	2	28.87	41	cellular biosynthetic process	all_reinforcing	40
GO:0034654	0.0053	2.1	12.88	22	nucleobase-containing compound biosynthetic process	all_reinforcing	40
GO:0008610	0.0063	3.1	3.47	9	lipid biosynthetic process	all_reinforcing	40
GO:0006696	0.0064	5	1.23	5	ergosterol biosynthetic process	all_reinforcing	40
GO:0015804	0.0087	9.3	0.46	3	neutral amino acid transport	all_reinforcing	40
GO:0015074	6.3E-08	12.3	1.5	11	DNA integration	all_reinforcing	45
GO:0090502	3.7E-07	9.9	1.7	11	RNA phosphodiester bond hydrolysis; endonucleolytic	all_reinforcing	45
GO:0006278	3.7E-07	9.9	1.7	11	RNA-dependent DNA biosynthetic process	all_reinforcing	45
GO:0032197	6.3E-07	9.3	1.8	11	transposition; RNA-mediated	all_reinforcing	45
GO:0006310	2.8E-05	5.3	3.1	12	DNA recombination	all_reinforcing	45
GO:0090305	7.6E-05	5.1	2.9	11	nucleic acid phosphodiester bond hydrolysis	all_reinforcing	45
GO:0044249	0.00039	2.5	22.9	36	cellular biosynthetic process	all_reinforcing	45
GO:1901576	0.0005	2.5	23.2	36	organic substance biosynthetic process	all_reinforcing	45
GO:1901360	0.0018	2.2	20.7	32	organic cyclic compound metabolic process	all_reinforcing	45
GO:0034654	0.0024	2.4	10	19	nucleobase-containing compound biosynthetic process	all_reinforcing	45
GO:0006696	0.0029	6.2	1	5	ergosterol biosynthetic process	all_reinforcing	45
GO:0016128	0.0048	5.4	1.2	5	phytosteroid metabolic process	all_reinforcing	45
GO:0044107	0.0056	5.2	1.2	5	cellular alcohol metabolic process	all_reinforcing	45
GO:1902652	0.0056	5.2	1.2	5	secondary alcohol metabolic process	all_reinforcing	45
GO:0055114	0.0075	2.4	7.1	14	oxidation-reduction process	all_reinforcing	45
GO:0006694	0.0075	4.8	1.3	5	steroid biosynthetic process	all_reinforcing	45
GO:0046165	0.0087	4.6	1.3	5	alcohol biosynthetic process	all_reinforcing	45
GO:0006644	0.0099	4.4	1.4	5	phospholipid metabolic process	all_reinforcing	45
GO:0015074	2.5E-05	8.8	1.32	8	DNA integration	all_reinforcing	50
GO:0006278	5.6E-05	7.7	1.47	8	RNA-dependent DNA biosynthetic process	all_reinforcing	50
GO:0090502	6.8E-05	7.5	1.5	8	RNA phosphodiester bond hydrolysis; endonucleolytic	all_reinforcing	50
GO:0032197	0.00012	6.8	1.61	8	transposition; RNA-mediated	all_reinforcing	50
GO:0006310	0.00052	4.7	2.49	9	DNA recombination	all_reinforcing	50
GO:0006696	0.0011	8.1	0.84	5	ergosterol biosynthetic process	all_reinforcing	50
GO:0016128	0.002	6.9	0.95	5	phytosteroid metabolic process	all_reinforcing	50
GO:0090305	0.002	4.2	2.42	8	nucleic acid phosphodiester bond hydrolysis	all_reinforcing	50
GO:0044107	0.0023	6.6	0.99	5	cellular alcohol metabolic process	all_reinforcing	50
GO:1902652	0.0023	6.6	0.99	5	secondary alcohol metabolic process	all_reinforcing	50
GO:0006694	0.0033	6	1.06	5	steroid biosynthetic process	all_reinforcing	50
GO:0046165	0.0038	5.8	1.1	5	alcohol biosynthetic process	all_reinforcing	50
GO:0016125	0.0075	4.8	1.28	5	sterol metabolic process	all_reinforcing	50
GO:1901362	0.0094	2.4	8.21	15	organic cyclic compound biosynthetic process	all_reinforcing	50

Abbreviations: "GOBPID": gene ontology biological process ID, "ExpCount": expected number of genes for a given GOBPID, "LOD": LOD score threshold for including a gene with a cis / trans eQTL pair.

Because the results of our GO enrichment could be affected by the reference gene set (the list of all genes with at least one cis and one trans eQTL), we devised a complementary strategy to test for lineage-specific selection on UPS gene expression. Using the same set of BY / RM eQTLs, we applied the sign test to the sets of UPS genes, proteasome genes, ubiquitin system genes, E3 ligases, and proteasome chaperone genes. We did not detect lineage-specific selection in any of these gene sets (Author response table 6).

**Author response table 6. sa2table6:** Results of the Sign Test for Lineage-Specific Selection Applied to UPS Gene BY / RM eQTLs.

LOD	n_eQTLs	n_genes	n_pairs	reinf_by_up	reinf_rm_up	oppos_by_up	oppos_rm_up	excess_reinforcing_pairs	chi_sq_p	gene_set
2.4	1128	186	95	16	30	29	20	–4.21	0.815	all_UPS_genes
5	587	176	72	13	23	23	13	0	1	all_UPS_genes
10	250	142	29	2	12	8	7	–4.41	0.428	all_UPS_genes
15	162	105	19	1	8	7	3	–2.74	0.619	all_UPS_genes
20	115	85	10	0	4	4	2	–3.2	0.472	all_UPS_genes
25	90	72	8	0	3	3	2	-3	0.475	all_UPS_genes
30	63	56	5	0	2	2	1	–1.6	1	all_UPS_genes
35	52	47	4	0	1	2	1	-2	1	all_UPS_genes
40	37	34	3	0	1	2	0	0	NaN	all_UPS_genes
45	32	30	2	0	1	1	0	0	NaN	all_UPS_genes
2.4	242	33	14	3	3	2	6	–0.857	1	proteasome_genes
5	146	32	12	3	2	2	5	–1.33	1	proteasome_genes
10	62	30	4	0	0	0	4	0	NaN	proteasome_genes
15	32	19	2	0	0	0	2	0	NaN	proteasome_genes
20	18	14	1	0	0	0	1	0	NaN	proteasome_genes
25	11	9	1	0	0	0	1	0	NaN	proteasome_genes
2.4	829	145	76	13	23	25	15	-4	0.812	ubiquitin_system_genes
5	410	137	57	10	18	20	9	0	1	ubiquitin_system_genes
10	173	105	24	2	11	7	4	-1	1	ubiquitin_system_genes
15	118	80	17	1	7	7	2	–1.65	1	ubiquitin_system_genes
20	88	65	8	0	3	4	1	-2	1	ubiquitin_system_genes
25	73	58	6	0	2	3	1	-2	1	ubiquitin_system_genes
30	54	48	4	0	1	2	1	-2	1	ubiquitin_system_genes
35	44	40	3	0	0	2	1	–2.67	0.324	ubiquitin_system_genes
40	31	29	2	0	0	2	0	0	NaN	ubiquitin_system_genes
45	26	25	1	0	0	1	0	0	NaN	ubiquitin_system_genes
2.4	618	111	56	11	16	19	10	-1	1	E3_ligase_genes
5	300	103	39	8	12	14	5	2.67	0.909	E3_ligase_genes
10	129	79	15	2	7	4	2	1.6	1	E3_ligase_genes
15	83	56	9	1	4	4	0	1.78	1	E3_ligase_genes
20	64	46	5	0	2	3	0	0	NaN	E3_ligase_genes
25	49	39	3	0	1	2	0	0	NaN	E3_ligase_genes
30	37	33	3	0	1	2	0	0	NaN	E3_ligase_genes
35	30	28	2	0	0	2	0	0	NaN	E3_ligase_genes
40	22	20	2	0	0	2	0	0	NaN	E3_ligase_genes
45	18	17	1	0	0	1	0	0	NaN	E3_ligase_genes
2.4	57	9	6	0	4	2	0	0	NaN	proteasome_chaperone_genes
5	31	8	4	0	3	1	0	0	NaN	proteasome_chaperone_genes
10	15	8	2	0	1	1	0	0	NaN	proteasome_chaperone_genes
15	12	7	1	0	1	0	0	0	NaN	proteasome_chaperone_genes
20	9	6	1	0	1	0	0	0	NaN	proteasome_chaperone_genes
25	6	5	1	0	1	0	0	0	NaN	proteasome_chaperone_genes
30	5	4	1	0	1	0	0	0	NaN	proteasome_chaperone_genes
35	4	3	1	0	1	0	0	0	NaN	proteasome_chaperone_genes
40	4	3	1	0	1	0	0	0	NaN	proteasome_chaperone_genes
45	4	3	1	0	1	0	0	0	NaN	proteasome_chaperone_genes

Abbreviations: “LOD”: LOD score threshold for calling cis / trans eQTL pairs, "n_eQTLs": number of eQTLs, "n_genes": number of genes with an eQTL, "n_pairs": number of cis / trans eQTL pairs, "reinf_BY_up": number of cis / trans eQTLs where the BY allele of the cis and trans eQTLs increases expression (reinforcing pairs), "reinf_RM_up": number of cis / trans eQTLs where the RM allele of the cis and trans eQTLs increases expression (reinforcing pairs), "oppos_BY_up": number of cis / trans eQTLs where the BY allele of the cis eQTL increases expression and the RM allele of the trans eQTL increases expression (opposing pairs), "oppos_RM_up": number of cis / trans eQTLs where the RM allele of the cis eQTL increases expression and the BY allele of the trans eQTL increases expression (opposing pairs), "excess reinforcing pairs" the number of excess reinforcing pairs, calculated as in Fraser et al., 2010, "chi_sq_p": p-value of the chi-square test for enrichment of reinforcing pairs.

Although we did not detect lineage-specific selection on global or UPS gene mRNA abundance, we note that these analyses do not detect lineage-specific selection on factors other than transcript abundance. For example, lineage-specific selection may occur through causal missense variants, such as the ones we identified here. For example, the *DOA10* and *NTA1* genes each contain multiple causal missense variants that alter UPS activity, but neither gene's transcript abundance is affected by a local eQTL. Integration of the effects of missense variants with those that alter gene expression in selection tests remains an interesting avenue for future work.

We note that the significant excess of UPS QTLs at which the RM allele increases UPS activity (pg. 6, para. 2, line 170) provides evidence that N-end Rule activity could have been subject to lineage-specific selection. To test whether this result was due to a strong enrichment of RM alleles at certain individual reporters, we applied this analysis to the sets of QTLs obtained for the 20 individual N-degrons. We did not detect an enrichment of any of the individual N-degrons, suggesting that the enrichment of RM QTLs that increase UPS activity is a general effect (Author response table 7).

**Author response table 7. sa2table7:** Results of Binomial Enrichment Test Applied to QTLs for Individual N-degrons.

N-degron	N_QTLs	N_RM_up	N_BY_up	p_value
Ala	15	10	5	0.30
Arg	7	4	3	1.00
Asn	7	3	4	1.00
Asp	8	5	3	0.73
Cys	9	4	5	1.00
Gln	3	2	1	1.00
Glu	4	2	2	1.00
Gly	9	5	4	1.00
His	9	6	3	0.51
Ile	1	1	0	1.00
Leu	2	1	1	1.00
Lys	7	5	2	0.45
Met	4	1	3	0.63
Phe	10	6	4	0.75
Pro	13	8	5	0.58
Ser	9	6	3	0.51
Thr	12	8	4	0.39
Trp	6	4	2	0.69
Tyr	7	4	3	1.00
Val	7	4	3	1.00

Abbreviations: "N_QTLs": Number of QTLs for the indicated N-degron, "N_RM_up": Number of QTLs where the RM allele increases UPS activity for the indicated N-degron, "N_BY_up": Number of QTLs where the BY allele increases UPS activity for the indicated N-degron. "p_value": p-value of the binomial test for the set of QTLs for the indicated N-degron.

The paper reports 149 loci, but if I am reading this correctly it looks like this may double-count shared loci. It would be nice to discuss a bit more about likely sharing (ie pleiotropy) of the hits. (It's also a bit hard to see from 2B which of these are likely shared.)

The reviewer is correct. As described in our response to Reviewer 1, the revised manuscript now indicates that the 149 instances of QTL detection correspond to 35 distinct QTL regions (pg. 6, para. 3, line 185). Figure 2—figure supplement 2 provides detailed information on these regions, including which reporters each region affects. The revised manuscript includes an extended discussion of pleiotropy among the set of N-end Rule QTLs (pg. 6, para. 3).

L143: it should be possible to make this statement quantitative.

As noted in our response to Reviewer 1, it is not readily possible to calculate the amount of variance explained by each QTL due to the pooled nature of our bulk segregant analysis QTL mapping method. Accordingly, we have removed the statement mentioned by the reviewer from the revised Results section.

Para 303, 312: It seems likely that you may be looking for an effect that is small at individual genes but widespread. I would suggest looking more carefully at the overall distribution: eg in the protein data is there an upward shift in the mean? You could also use a more sophisticated method like ashR to study the distribution of changes.

We appreciate the reviewer’s suggestion, which we suspect may have been prompted by our wording of the following sentence from the Results section (page 23, paragraph 2, lines 307-310 of the original manuscript):

“This result is consistent with recent observations that suggest that altering UBR1 expression exerts broad effects on protein degradation or related processes controlling protein abundance and that protein sequences, rather than function or subcellular localization, are the primary determinants of degradation rates.”

Our intent was to convey that substrates of E3 ligases such as Ubr1 are more likely to share sequence features than they are to share a function or subcellular localization. In other words, E3 ligases typically target sets of proteins that are functionally diverse but that share common sequence features, e.g., an N-degron. The wording of this sentence may have unintentionally suggested that the causal *UBR1* promoter variant should affect the abundance of many proteins. However, E3 ligases influence the abundance of distinct sets of dozens to several hundred proteins (Kong et al., 2021, Christiano et al., 2020). The moderate effect of the causal *UBR1* promoter variant on *UBR1* expression is, therefore, not expected to create widespread effects on the proteome but instead to affect a small subset of Ubr1 substrates and related proteins. We have revised this section to more clearly articulate these ideas. We have also re-analysed our data as described below.

The causal *UBR1* variant significantly altered the abundance of 39 of 3,047 detected proteins at a 0.1 false discovery rate (FDR) threshold. Following the reviewer’s suggestion, we computed the overall median log_2_ fold change and found that it was -0.012 for all 3,047 detected proteins (that is, very close to zero) and 0.37 for the set of 39 differentially abundant proteins (that is, an average increase for proteins with significantly different abundance). The number of differentially abundant proteins, the average upward shift in log2 fold change for differentially abundant proteins, and the significant fraction of differentially abundant proteins exhibiting increased abundance (28 / 39, 72%, binomial p = 9.5e-3) are all consistent with the causal variant’s moderate effect on *UBR1* expression.

The causal variant also did not have widespread effects on the transcriptome (median log_2_ fold change = -0.0024 [again, very close to zero]). As reported in our initial submission, 78 genes were differentially expressed at an FDR of 0.1. For these genes, the overall effect of the causal BY allele, which decreases *UBR1* expression, was to decrease transcript abundance (median log_2_ fold change = -0.18, 60 / 78 decreased expression [77%, binomial p = 2e-6]).

Following the reviewer’s suggestion, we used ashr to explore how using the false sign rate (FSR, Stephens, 2017; https://doi.org/10.1093/biostatistics/kxw041) to call differentially expressed genes influenced our analysis of the causal *UBR1* variant’s effects. Using an FSR threshold of 0.1, we detected 86 genes with altered mRNA transcript abundance. For these genes, the BY allele tended to decrease transcript abundance (median log_2_ fold change = -0.2, 65 / 86 [76%, binomial p = 2e-6]). All differentially expressed genes detected using the FDR were also detected using the FSR. Eight additional differentially expressed genes were detected using the FSR, but not the FDR. Each of these eight genes had lower absolute log_2_ fold changes than the 78 differentially expressed genes detected using the FDR. Therefore, we conclude that while the FSR is slightly more permissive, the FDR and FSR both capture the same global patterns of effects in our RNA-seq data.

In contrast to these consistent results in RNA-seq, ashr produced different results than the FDR when applied to our proteomics data. As reported in our initial submission, a 0.1 FDR threshold applied to abundance differences reported by the Proteome Discoverer software results in 39 differentially abundant proteins. The derived BY *UBR1* allele increased the abundance of 28 of these 39 (72%, binomial p = 9.5e-3) differentially abundant proteins. At a 0.1 FSR threshold, we identified 56 differentially abundant proteins, with the BY allele increasing the abundance of 13 / 56 (23%, binomial p = 7.3e-5).This difference arises from the fact that there are considerable discrepancies among the sets of differentially abundant proteins called by FDR and FSR. We note that of the 39 differentially abundant proteins reported by Proteome Discoverer, 12 were estimated to have large absolute fold changes (>= 0.5, the 99th percentile observed in our data) but were not reported as significant by FSR (Author response image 1). Notably, these 12 genes included the known Ubr1-regulated proteins Tma10 and Adh2 (Kong et al., 2021, Christiano et al., 2020). For 8 of these 12 proteins, the BY allele increased protein abundance. These 12 proteins had relatively high standard errors and, as expected, their fold changes were considerably lower following the adaptive shrinkage that is part of ashr (Author response image 1). Second, ashr identified an additional 48 differentially abundant proteins. The BY allele decreases the abundance of 41 of these 48 proteins (Author response image 1).

Given that ashr strongly depends on correctly estimated standard errors, its application to mass-spectrometry data with a small sample size could be potentially problematic in a manner that does not seem to apply to RNA-seq, which, particularly at the very high sequencing depth used here, produces more accurate estimates of the standard error. We have therefore elected to continue to use the FDR applied to p-values reported by Proteome Discoverer to call differentially expressed genes at the protein and RNA levels. We note that our general conclusion that the causal UBR1 promoter variant affects the expression of dozens of genes at the protein and RNA levels remains unchanged irrespective of the method used to call differentially expressed genes.

**Author response image 1. sa2fig1:** 

For Figure 5 you could also consider adding information about global patterns in means and correlations, which are difficult to read from these plots.

The revised Figure 5 contains the median log_2_ fold changes for our proteomics and RNA-seq data (pg. 14). The revised Results section reports the correlation in log_2_ fold change for genes detected in both our proteomics (pg. 13, para. 4, line 362) and RNA-seq data (pg. 13, para. 5, line 377).